# Flow Factorized Representation Learning

**Yue Song**[1,2], **T. Anderson Keller**[2, 3], **Nicu Sebe**[1], and **Max Welling**[2,3]

[1]Department of Information Engineering and Computer Science, University of Trento, Italy
[2]Amsterdam Machine Learning Lab, University of Amsterdam, the Netherlands
[3]UvA-Bosch Delta Lab, University of Amsterdam, the Netherlands
`yue.song@unitn.it`

## Abstract

A prominent goal of representation learning research is to achieve representations which are factorized in a useful manner with respect to the ground truth factors of variation. The fields of disentangled and equivariant representation learning have approached this ideal from a range of complimentary perspectives; however, to date, most approaches have proven to either be ill-specified or insufficiently flexible to effectively separate all realistic factors of interest in a learned latent space. In this work, we propose an alternative viewpoint on such structured representation learning which we call Flow Factorized Representation Learning, and demonstrate it to learn both more efficient and more usefully structured representations than existing frameworks. Specifically, we introduce a generative model which specifies a distinct set of latent probability paths that define different input transformations. Each latent flow is generated by the gradient field of a learned potential following dynamic optimal transport. Our novel setup brings new understandings to both *disentanglement* and *equivariance*. We show that our model achieves higher likelihoods on standard representation learning benchmarks while simultaneously being closer to approximately equivariant models. Furthermore, we demonstrate that the transformations learned by our model are flexibly composable and can also extrapolate to new data, implying a degree of robustness and generalizability approaching the ultimate goal of usefully factorized representation learning.

## 1 Introduction

Developing models which learn useful representations of data has become an increasingly important focus in the machine learning community [5, 55]. For example, Large Language Models such as GPT [9] rely on an extensive pre-training phase to learn valuable representations, enabling quick finetuning on a diversity of tasks. However, a precise definition of what makes an ideal representation is still debated. One line of work has focused on 'disentanglement' of the underlying ground truth generative factors [5, 35, 13]. In general, the definition of 'disentanglement' often refers to learning and controlling statistically independent factors of variation [5, 36]. Over the years, many disentanglement methods have been proposed, including axis-aligned single-dimensional manipulation [35, 13], linear multi-dimensional traversals [78, 77, 90, 66], and, more recently, dynamic non-linear vector-based traversals [84, 79]. Although these methods have been met with significant success (and even linked to single-neuron brain activity [37, 91]), there are known theoretical limitations which make them ill-specified, including the presence of topological defects [7]. This has limited their deployment beyond toy settings.

Another line of work has focused on developing representations which respect symmetries of the underlying data in their output space [15, 36]. Specifically, equivariant representations are those for which the output transforms in a known predictable way for a given input transformation.

37th Conference on Neural Information Processing Systems (NeurIPS 2023).

They can be seen to share many similarities with disentangled representations since an object undergoing a transformation which preserves its identity can be called a symmetry transformation [36]. Compared with disentanglement methods, equivariant networks are much more strictly defined, allowing for significantly greater control and theoretical guarantees with respect to the learned transformation [16, 50, 73, 20, 39]. However, this restriction also limits the types of transformations to which they may be applied. For example, currently only group transformations are supported, limiting real-world applicability. To avoid this caveat, some recent attempts propose to learn general but approximate equivariance from disentangled representations [49, 45, 79].

In this work, we provide an alternative viewpoint at the intersection of these two fields of work which we call Flow Factorized Representation Learning. Fig. 1 depicts the high-level illustration of our method. Given $k$ different transformations $p_k(\boldsymbol{x}_t|\boldsymbol{x}_0)$ in the input space, we have the corresponding latent probabilistic path $\int_{\boldsymbol{z}_0,\boldsymbol{z}_t} q(\boldsymbol{z}_0|\boldsymbol{x}_0)q_k(\boldsymbol{z}_t|\boldsymbol{z}_0)p(\boldsymbol{x}_t|\boldsymbol{z}_t)$ for each of the transformations. Each latent flow path $q_k(\boldsymbol{z}_t|\boldsymbol{z}_0)$ is generated by the gradient field of some learned potentials $\nabla u^k$ following fluid mechanical dynamic Optimal Transport (OT) [4]. Our framework allows for novel understandings of both *disentanglement* and *equivariance*. The definition of disentanglement refers to the distinct set of tangent directions $\nabla u^k$ that follow the OT paths to generate latent flows for modeling different factors of variation. The concept of equivariance in our case means that the two probabilistic paths, *i.e.,* $p_k(\boldsymbol{x}_t|\boldsymbol{x}_0)$ in the image space and $\int_{\boldsymbol{z}_0,\boldsymbol{z}_t} q(\boldsymbol{z}_0|\boldsymbol{x}_0)q_k(\boldsymbol{z}_t|\boldsymbol{z}_0)p(\boldsymbol{x}_t|\boldsymbol{z}_t)$ in the latent space, would eventually result in the same distribution of transformed data.

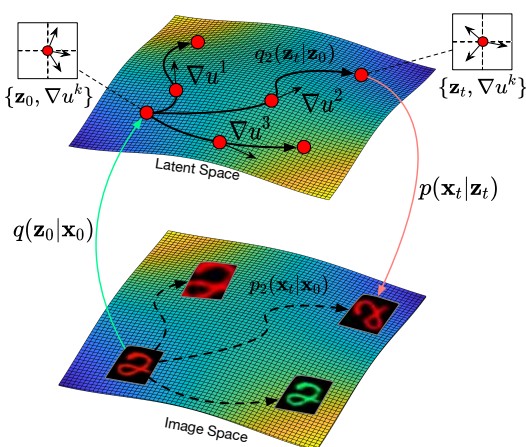

Figure 1: Illustration of our flow factorized representation learning: at each point in the latent space we have a distinct set of tangent directions $\nabla u^k$ which define different transformations we would like to model in the image space. For each path, the latent sample evolves to the target on the potential landscape following dynamic optimal transport.

We build a formal generative model of sequences and integrate the above latent probability evolution as condition updates of the factorized sequence distribution. Based on the continuity equation, we derive a proper flow of probability density for the time evolution of both the prior and posterior. To perform inference, we approximate the true posterior of latent variables and train the parameters as a Variational Autoencoder (VAE) [47]. When the transformation type $k$ is not observed (*i.e.,* available as a label), we treat $k$ as another latent variable and incorporate its posterior into our framework by learning it from sequences. Extensive experiments and thorough analyses have been conducted to show the effectiveness of our method. For example, we demonstrate empirically that our representations are usefully factorized, allowing flexible composability and generalization to new datasets. Furthermore, we show that our methods are also approximately equivariant by demonstrating that they commute with input transformations through the learned latent flows. Ultimately, we see these factors combine to yield the highest likelihood on the test set in each setting. Code is publicly available at `https://github.com/KingJamesSong/latent-flow`.

## 2 The generative model

In this section, we first introduce our generative model of sequences and then describe how we perform inference over the latent variables of this model in the next section.

### 2.1 Flow factorized sequence distributions

The model in this work defines a distribution over sequences of observed variables. We further factorize this distribution into $k$ distinct components by assuming that each observed sequence is generated by one of the $k$ separate flows of probability mass in latent space. Since in this work we

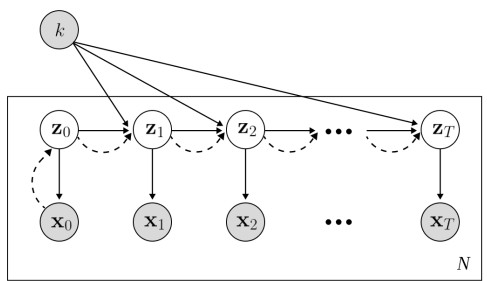 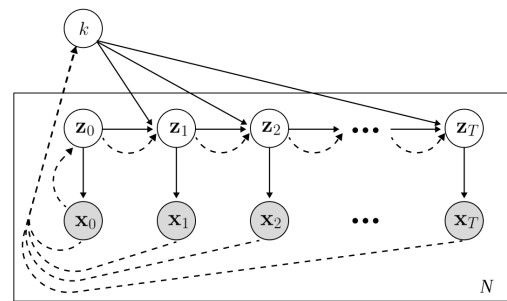

Figure 2: Depiction of our model in plate notation. (Left) Supervised, (Right) Weakly-supervised. White nodes denote latent variables, shaded nodes denote observed variables, solid lines denote the generative model, and dashed lines denote the approximate posterior. We see, as in a standard VAE framework, our model approximates the initial one-step posterior $p(\boldsymbol{z}_0|\boldsymbol{x}_0)$, but additionally approximates the conditional transition distribution $p(\boldsymbol{z}_t|\boldsymbol{z}_{t-1}, k)$ through dynamic optimal transport over a potential landscape.

model discrete sequences of observations $\bar{\boldsymbol{x}} = \{\boldsymbol{x}_0, \boldsymbol{x}_1 \ldots, \boldsymbol{x}_T\}$, we aim to define a joint distribution with a similarly discrete sequence of latent variables $\bar{\boldsymbol{z}} = \{\boldsymbol{z}_0, \boldsymbol{z}_1 \ldots, \boldsymbol{z}_T\}$, and a categorical random variable $k$ describing the sequence type (observed or unobserved). Explicitly, we assert the following factorization of the joint distribution over $T$ timesteps:

$$p(\bar{\boldsymbol{x}}, \bar{\boldsymbol{z}}, k) = p(k)p(\boldsymbol{z}_0)p(\boldsymbol{x}_0|\boldsymbol{z}_0) \prod_{t=1}^{T} p(\boldsymbol{z}_t|\boldsymbol{z}_{t-1}, k)p(\boldsymbol{x}_t|\boldsymbol{z}_t). \tag{1}$$

Here $p(k)$ is a categorical distribution defining the transformation type, $p(\boldsymbol{x}_t|\boldsymbol{z}_t)$ asserts a mapping from latents to observations with Gaussian noise, and $p(\boldsymbol{z}_0) = \mathcal{N}(0, 1)$. A plate diagram of this model is depicted through the solid lines in Fig. 2.

## 2.2 Prior time evolution

To enforce that the time dynamics of the sequence define a proper flow of probability density, we compute the conditional update $p(\boldsymbol{z}_t|\boldsymbol{z}_{t-1}, k)$ from the continuous form of the continuity equation: $\partial_t p(\boldsymbol{z}) = -\nabla \cdot (p(\boldsymbol{z})\nabla\psi^k(\boldsymbol{z}))$, where $\psi^k(\boldsymbol{z})$ is the $k$'th potential function which advects the density $p(\boldsymbol{z})$ through the induced velocity field $\nabla\psi^k(\boldsymbol{z})$. Considering the discrete particle evolution corresponding to this density evolution, $\boldsymbol{z}_t = f(\boldsymbol{z}_{t-1}, k) = \boldsymbol{z}_{t-1} + \nabla_z \psi^k(\boldsymbol{z}_{t-1})$, we see that we can derive the conditional update from the continuous change of variables formula [69, 11]:

$$p(\boldsymbol{z}_t|\boldsymbol{z}_{t-1}, k) = p(\boldsymbol{z}_{t-1})\left|\frac{df(\boldsymbol{z}_{t-1}, k)}{d\boldsymbol{z}_{t-1}}\right|^{-1} \tag{2}$$

In this setting, we see that the choice of $\psi$ ultimately determines the prior on the transition probability in our model. As a minimally informative prior for random trajectories, we use a diffusion equation achieved by simply taking $\psi^k = -D_k \log p(\boldsymbol{z}_t)$. Then according to the continuity equation, the prior evolves as:

$$\partial_t p(\boldsymbol{z}_t) = -\nabla \cdot \left(p(\boldsymbol{z}_t)\nabla\psi\right) = D_k \nabla^2 p(\boldsymbol{z}_t) \tag{3}$$

where $D_k$ is a constant coefficient that does not change over time. The density evolution of the prior distribution thus follows a constant diffusion process. We set $D_k$ as a learnable parameter which is distinct for each $k$.

## 3 Flow factorized variational autoencoders

To perform inference over the unobserved variables in our model, we propose to use a variational approximation to the true posterior, and train the parameters of the model as a VAE. To do this, we parameterize an approximate posterior for $p(\boldsymbol{z}_0|\boldsymbol{x}_0)$, and additionally parameterize a set of $K$

functions $u^k(z)$ to approximate the true latent potentials $\psi^*$. First, we will describe how we do this in the setting where the categorical random variable $k$ is observed (which we call the supervised setting), then we will describe the model when $k$ is also latent and thus additionally inferred (which we call the weakly supervised setting).

## 3.1 Inference with observed $k$ (supervised)

When $k$ is observed, we define our approximate posterior to factorize as follows:

$$q(\bar{z}|\bar{x}, k) = q(z_0|x_0) \prod_{t=1}^{T} q(z_t|z_{t-1}, k) \tag{4}$$

We see that, in effect, our approximate posterior only considers information from element $x_0$; however, combined with supervision in the form of $k$, we find this is sufficient for the posterior to be able to accurately model full latent sequences. In the limitations section we discuss how the posterior could be changed to include all elements $\{x_t\}_0^T$ in future work.

Combing Eq. (4) with Eq. (1), we can derive the following lower bound to model evidence (ELBO):

$$
\begin{aligned}
\log p(\bar{x}|k) &= \mathbb{E}_{q_\theta(\bar{z}|\bar{x},k)} \left[ \log \frac{p(\bar{x}, \bar{z}|k)}{q(\bar{z}|\bar{x}, k)} \frac{q(\bar{z}|\bar{x}, k)}{p(\bar{z}|\bar{x}, k)} \right] \\
&\geq \mathbb{E}_{q_\theta(\bar{z}|\bar{x},k)} \left[ \log \frac{p(\bar{x}|\bar{z}, k)p(\bar{z}|k)}{q(\bar{z}|\bar{x}, k)} \right] \\
&= \mathbb{E}_{q_\theta(\bar{z}|\bar{x},k)} \left[ \log p(\bar{x}|\bar{z}, k) \right] + \mathbb{E}_{q_\theta(\bar{z}|\bar{x},k)} \left[ \log \frac{p(\bar{z}|k)}{q(\bar{z}|\bar{x}, k)} \right]
\end{aligned}
\tag{5}
$$

Substituting and simplifying, Eq. (5) can be re-written as

$$
\begin{aligned}
\log p(\bar{x}|k) &\geq \sum_{t=0}^{T} \mathbb{E}_{q_\theta(\bar{z}|k)} \left[ \log p(x_t|z_t, k) \right] - \mathbb{E}_{q_\theta(\bar{z}|k)} \left[ D_{\text{KL}} \left[ q_\theta(z_0|x_0) || p(z_0) \right] \right] \\
&\quad - \sum_{t=1}^{T} \mathbb{E}_{q_\theta(\bar{z}|k)} \left[ D_{\text{KL}} \left[ q_\theta(z_t|z_{t-1}, k) || p(z_t|z_{t-1}, k) \right] \right]
\end{aligned}
\tag{6}
$$

We thus see that we have an objective very similar to that of a traditional VAE, except that our posterior and our prior now both have a time evolution defined by the conditional distributions.

## 3.2 Inference with latent $k$ (weakly supervised)

When $k$ is not observed, we can treat it as another latent variable, and simultaneously perform inference over it in addition to the sequential latent $\bar{z}$. To achieve this, we define our approximate posterior and instead factorize it as

$$q(\bar{z}, k|\bar{x}) = q(k|\bar{x}) q(z_0|x_0) \prod_{t=1}^{T} q(z_t|z_{t-1}, k) \tag{7}$$

Following a similar procedure as in the supervised setting, we derive the new ELBO as

$$
\begin{aligned}
\log p(\bar{x}) &= \mathbb{E}_{q_\theta(\bar{z},k|\bar{x})} \left[ \log \frac{p(\bar{x}, \bar{z}, k)}{q(\bar{z}, k|\bar{x})} \frac{q(\bar{z}, k|\bar{x})}{p(\bar{z}, k|\bar{x})} \right] \\
&\geq \mathbb{E}_{q_\theta(\bar{z},k|\bar{x})} \left[ \log \frac{p(\bar{x}|\bar{z}, k)p(\bar{z}|k)}{q(\bar{z}|\bar{x}, k)} \frac{p(k)}{q(k|\bar{x})} \right] \\
&= \mathbb{E}_{q_\theta(\bar{z},k|\bar{x})} \left[ \log p(\bar{x}|\bar{z}, k) \right] + \mathbb{E}_{q_\theta(\bar{z},k|\bar{x})} \left[ \log \frac{p(\bar{z}|k)}{q(\bar{z}|\bar{x}, k)} \right] + \mathbb{E}_{q_\gamma(k|\bar{x})} \left[ \log \frac{p(k)}{q(k|\bar{x})} \right]
\end{aligned}
\tag{8}
$$

We see that, compared with Eq. (5), only one additional KL divergence term $D_{\text{KL}} \left[ q_\gamma(k|\bar{x}) || p(k) \right]$ is added. The prior $p(k)$ is set to follow a categorical distribution, and we apply the `Gumbel-SoftMax` trick [43] to allow for categorical re-parameterization and sampling of $q_\gamma(k|\bar{x})$.

## 3.3 Posterior time evolution

As noted, to approximate the true generative model which has some unknown latent potentials $\psi^k$, we propose to parameterize a set of potentials as $u^k(\boldsymbol{z}, t) = \texttt{MLP}([\boldsymbol{z}; t])$ and train them through the ELBOs above. Again, we use the continuity equation to define the time evolution of the posterior, and thus we can derive the conditional time update $q(\boldsymbol{z}_t|\boldsymbol{z}_{t-1}, k)$ through the change of variables formula. Given the function of the sample evolution $\boldsymbol{z}_t = g(\boldsymbol{z}_{t-1}, k) = \boldsymbol{z}_{t-1} + \nabla_{\boldsymbol{z}} u^k$, we have:

$$q(\boldsymbol{z}_t|\boldsymbol{z}_{t-1}, k) = q(\boldsymbol{z}_{t-1}) \left| \frac{dg(\boldsymbol{z}_{t-1}, k)}{d\boldsymbol{z}_{t-1}} \right|^{-1} \tag{9}$$

Converting the above continuous equation to the discrete setting and taking the logarithm of both sides gives the normalizing-flow-like density evolution of our posterior:

$$\log q(\boldsymbol{z}_t|\boldsymbol{z}_{t-1}, k) = \log q(\boldsymbol{z}_{t-1}) - \log|1 + \nabla_{\boldsymbol{z}}^2 u^k| \tag{10}$$

The above relation can be equivalently derived from the continuity equation (*i.e.,* $\partial_t q(\boldsymbol{z}) = -\nabla \cdot \left( q(\boldsymbol{z}) \nabla u^k \right)$). Notice that we only assume the initial posterior $q(\boldsymbol{z}_0|\boldsymbol{x}_0)$ follows a Gaussian distribution. For future timesteps, we do not pose any further assumptions and just let the density evolve according to the sample motion.

## 3.4 Ensuring optimal transport of the posterior flow

As an inductive bias, we would like each latent posterior flow to follow the OT path. To accomplish this, it is known that when the gradient $\nabla u^k$ satisfies certain PDEs, the evolution of the probability density can be seen to minimize the $L_2$ Wasserstein distance between the source distribution and the distribution of the target transformation. Specifically, we have:

**Theorem 1** (Benamou-Brenier Formula [4]). *For probability measures $\mu_0$ and $\mu_1$, the $L_2$ Wasserstein distance can be defined as*

$$W_2(\mu_0, \mu_1)^2 = \min_{\rho, v} \left\{ \int \int \frac{1}{2} \rho(x, t)|v(x, t)|^2 \, dx \, dt \right\} \tag{11}$$

*where the density $\rho$ and the velocity $v$ satisfy:*

$$\frac{d\,\rho(x, t)}{dt} = -\nabla \cdot (v(x, t)\rho(x, t)), \; v(x, t) = \nabla u(x, t) \tag{12}$$

The optimality condition of the velocity is given by the generalized Hamilton-Jacobi (HJ) equation (*i.e.,* $\partial_t u + 1/2||\nabla u||^2 \leq 0$). The detailed derivation is deferred to the supplementary. We thus encourage our potential to satisfy the HJ equation with an external driving force as

$$\frac{\partial}{\partial t} u^k(\boldsymbol{z}, t) + \frac{1}{2}||\nabla_{\boldsymbol{z}} u^k(\boldsymbol{z}, t)||^2 = f(\boldsymbol{z}, t) \;\; \text{subject to} \;\; f(\boldsymbol{z}, t) \leq 0 \tag{13}$$

Here we use another $\texttt{MLP}$ to parameterize the external force $f(\boldsymbol{z}, t)$ and realize the negativity constraint by setting $f(\boldsymbol{z}, t) = -\texttt{MLP}([z; t])^2$. Notice that here we take the external force as learnable MLPs simply because we would like to obtain a flexible negativity constraint. The MLP architecture is set the same for both $u(\boldsymbol{z}, t)$ and $f(\boldsymbol{z}, t)$. To achieve the PDE constraint, we impose a Physics-Informed Neural Network (PINN) [67] loss as

$$\mathcal{L}_{HJ} = \frac{1}{T} \sum_{t=1}^{T} \left( \frac{\partial}{\partial t} u^k(\boldsymbol{z}, t) + \frac{1}{2}||\nabla_{\boldsymbol{z}} u^k(\boldsymbol{z}, t)||^2 - f(\boldsymbol{z}, t) \right)^2 + ||\nabla u^k(\boldsymbol{z}_0, 0)||^2 \tag{14}$$

where the first term restricts the potential to obey the HJ equation, and the second term limits $u(\boldsymbol{z}_t, t)$ to return no update at $t{=}0$, therefore matching the initial condition.

# 4 Experiments

This section starts with the experimental setup, followed by the main qualitative and quantitative results, then proceeds to discussions about the generalization ability to different composability and unseen data, and ends with the results on complex real-world datasets.

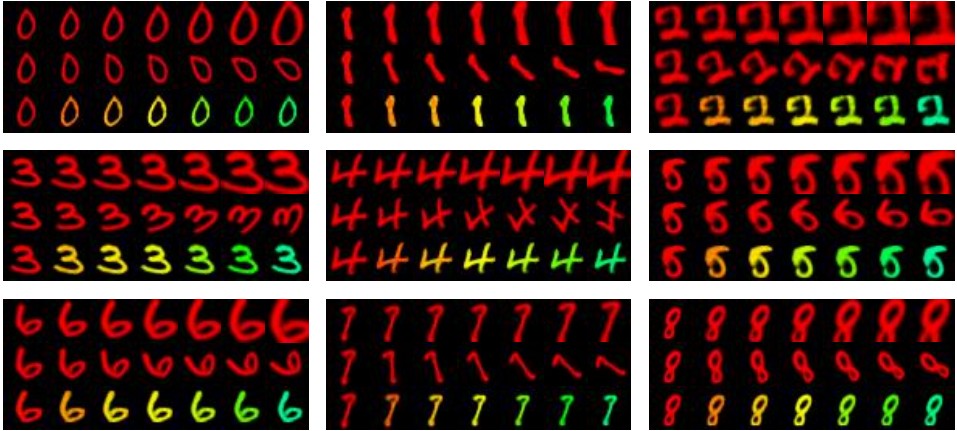

Figure 3: Exemplary latent evolution results of Scaling, Rotation, and Coloring on MNIST [54]. The top two rows are based on the supervised experiment, while the images of the bottom row are taken from the weakly-supervised setting of our experiment.

## 4.1 Setup

**Datasets.** We evaluate our method on two widely-used datasets in generative modeling, namely MNIST [54] and Shapes3D [10]. For MNIST [54], we manually construct three simple transformations including Scaling, Rotation, and Coloring. For Shapes3D [10], we use the self-contained four transformations that consist of Floor Hue, Wall Hue, Object Hue, and Scale.

Besides these two common benchmarks, we take a step further to apply our method on Falcol3D and Isaac3D [61], two complex *large-scale* and *real-world* datasets that contain sequences of different transformations. Falcol3D consists of indoor 3D scenes in different lighting conditions and viewpoints, while Isaac3D is a dataset of various robot arm movements in dynamic environments.

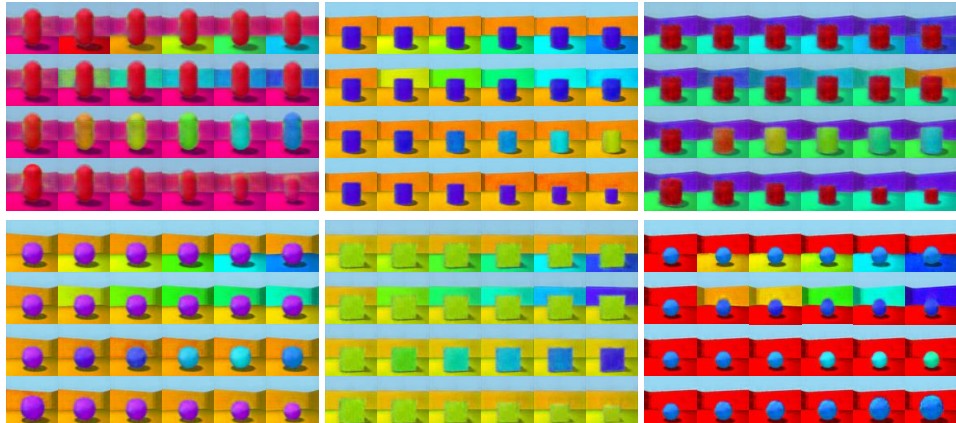

Figure 4: Exemplary latent flow results on Shapes3D [10]. The transformations from top to bottom are Floor Hue, Wall Hue, Object Hue, and Scale, respectively. The images of the top row are from the supervised experiment, while the bottom row is based on the weakly-supervised experiment.

**Baselines.** We mainly compare our method with SlowVAE [49] and Topographic VAE (TVAE) [45]. These two baselines could both achieve approximate equivariance. Specifically, TVAE introduces some learned latent operators, while SlowVAE enforces the Laplacian prior $p(z_t|z_{t-1}) = \prod \alpha\lambda/2\Gamma(1/\alpha) \exp\left(-\lambda|z_{t,i} - z_{t-1,i}|^\alpha\right)$ to sequential pairs. Within the disentanglement literature, our method is compared with the supervised PoFlow [79] which adopts a wave-like potential flow for sample evolution, and the unsupervised $\beta$-VAE [35] and FactorVAE [46] which encourage independence between single latent dimensions. Finally, the vanilla VAE is used as a controlled baseline.

| Methods | Supervision? | Equivariance Error (↓) | | | Log-likelihood (↑) |
|---|---|---|---|---|---|
| | | Scaling | Rotation | Coloring | |
| VAE [47] | No (✗) | 1275.31±1.89 | 1310.72±2.19 | 1368.92±2.33 | -2206.17±1.83 |
| $\beta$-VAE [35] | No (✗) | 741.58±4.57 | 751.32±5.24 | 808.16±5.03 | -2224.67±2.35 |
| FactorVAE [46] | No (✗) | 659.71±4.89 | 632.44±5.76 | 662.18±5.26 | -2209.33±2.47 |
| SlowVAE [49] | Weak (✓) | 461.59±5.37 | 447.46±5.46 | 398.12±4.83 | -2197.68±2.39 |
| TVAE [45] | Yes (✓) | 505.19±2.77 | 493.28±3.37 | 451.25±2.76 | -2181.13±1.87 |
| PoFlow [79] | Yes (✓) | 234.78±2.91 | 231.42±2.98 | 240.57±2.58 | -2145.03±2.01 |
| **Ours** | Yes (✓) | **185.42±2.35** | **153.54±3.10** | **158.57±2.95** | **-2112.45±1.57** |
| **Ours** | Weak (✓) | 193.84±2.47 | 157.16±3.24 | 165.19±2.78 | -2119.94±1.76 |

Table 1: Equivariance error $\mathcal{E}_k$ and log-likelihood $\log p(\boldsymbol{x}_t)$ on MNIST [54].

**Metrics.** We use the approximate equivariance error $\mathcal{E}_k$ and the log-likelihood of transformed data $\log p(\boldsymbol{x}_t)$ as the evaluation protocols. The equivariance error is defined as $\mathcal{E}_k = \sum_{t=1}^{T} |\boldsymbol{x}_t -$ $\texttt{Decode}(\boldsymbol{z}_t)|$ where $\boldsymbol{z}_t = \boldsymbol{z}_0 + \sum_{t=1}^{T} \nabla_{\boldsymbol{z}} u^k$. For TVAE, the latent operator is changed to $\texttt{Roll}(\boldsymbol{z}_0, t)$. For unsupervised disentanglement baselines [35, 46] and SlowVAE [49], we carefully select the latent dimension and tune the interpolation range to attain the traversal direction and range that correspond to the smallest equivariance error. Since the vanilla VAE does not have the corresponding learned transformation in the latent space, we simply set $\nabla_{\boldsymbol{z}} u^k = 0$ and take it as a lower-bound baseline. For all the methods, the results are reported based on 5 runs.

Notice that the above equivariance error is defined in the output space. Another reasonable evaluation metric is instead measuring error in the latent space as $\mathcal{E}_k = \sum_{t=1}^{T} |\texttt{Encode}(\boldsymbol{x}_t) - \boldsymbol{z}_t|$. We see the first evaluation method is more comprehensive as it further involves the decoder in the evaluation.

## 4.2 Main Results

**Qualitative results.** Fig. 3 and 4 display decoded images of the latent evolution on MNIST [54] and Shapes3D [10], respectively. On both datasets, our latent flow can perform the target transformation precisely during evolution while leaving other traits of the image unaffected. In particular, for the weakly-supervised setting, the decoded images (*i.e.,* the bottom rows of Fig. 3 and 4) can still reproduce the given transformations well and it is even hard to visually tell them apart from the generated images under the supervised setting. This demonstrates the effectiveness of the weakly-supervised setting of our method, and implies that qualitatively our latent flow is able to learn the sequence transformations well under both supervised and weakly-supervised settings.

| Methods | Supervision? | Equivariance Error (↓) | | | | Log-likelihood (↑) |
|---|---|---|---|---|---|---|
| | | Floor Hue | Wall Hue | Object Hue | Scale | |
| VAE [47] | No (✗) | 6924.63±8.92 | 7746.37±8.77 | 4383.54±9.26 | 2609.59±7.41 | -11784.69±4.87 |
| $\beta$-VAE [35] | No (✗) | 2243.95±12.48 | 2279.23±13.97 | 2188.73±12.61 | 2037.94±11.72 | -11924.83±5.64 |
| FactorVAE [46] | No (✗) | 1985.75±13.26 | 1876.41±11.93 | 1902.83±12.27 | 1657.32±11.05 | -11802.17±5.69 |
| SlowVAE [49] | Weak (✓) | 1247.36±12.49 | 1314.86±11.41 | 1102.28±12.17 | 1058.74±10.96 | -11475.89±5.74 |
| TVAE [45] | Yes (✓) | 1225.47±9.82 | 1246.32±9.54 | 1261.79±9.86 | 1142.01±9.37 | -11475.48±5.18 |
| PoFlow [79] | Yes (✓) | 885.46±10.37 | 916.71±10.49 | 912.48±9.86 | 924.39±10.05 | -11335.84±4.95 |
| **Ours** | Yes (✓) | **613.29±8.93** | **653.45±9.48** | **605.79±8.63** | **599.71±9.34** | **-11215.42±5.71** |
| **Ours** | Weak (✓) | 690.84±9.57 | 717.74±10.65 | 681.59±9.02 | 653.58±9.57 | -11279.61±5.89 |

Table 2: Equivariance error $\mathcal{E}_k$ and log-likelihood $\log p(\boldsymbol{x}_t)$ on Shapes3D [10].

**Quantitative results.** Tables 1 and 2 compare the equivariance error and the log-likelihood on MNIST [54] and Shapes3D [10], respectively. Our method learns the latent flows which model the transformations precisely, achieving the best performance across datasets under different supervision settings. Specifically, our method outperforms the previous best baseline by 69.74 on average in the equivariance error and by 32.58 in the log-likelihood on MNIST. The performance gain is also consistent on Shapes3D: our method surpasses the second-best baseline by 291.70 in the average equivariance error and by 120.42 in the log-likelihood. In the weakly-supervised setting, our method also achieves very competitive performance, falling behind that of the supervised setting in the average equivariance error slightly by 6.22 on MNIST and by 67.88 on Shapes3D.

## 4.3 Discussion

**Extrapolation: switching transformations.** In Fig. 5 we demonstrate that, empowered by our method, it is possible to switch latent transformation categories mid-way through the latent evolution

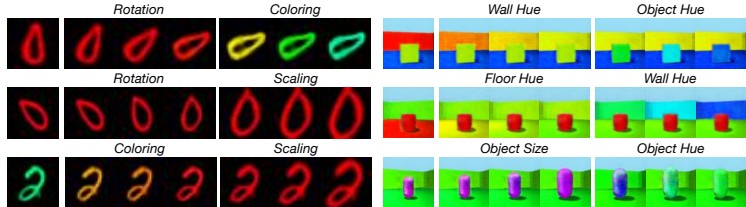

Figure 5: Exemplary visualization of switching transformations during the latent sample evolution.

and maintain coherence. That is, we perform $z_t = z_{t-1} + \nabla_z u^k$ for $t \leq T/2$ and then change to $z_t = z_{t-1} + \nabla_z u^j$ where $j \neq k$ for $t > T/2$. As can be seen, the factor of variation immediately changes after the transformation type is switched. Moreover, the transition phase is smooth and no other attributes of the image are influenced.

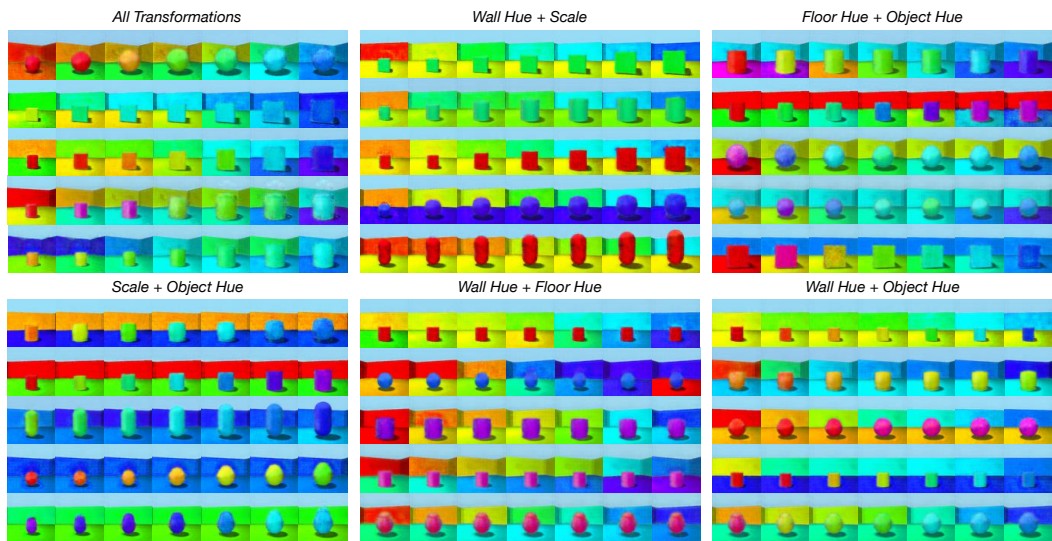

Figure 6: Examples of combining different transformations simultaneously during the latent evolution.

**Extrapolation: superposing transformations.** Besides switching transformations, our method also supports applying different transformations simultaneously, *i.e.,* consistently performing $z_t = z_{t-1} + \sum_k^K \nabla_z u^k$ during the latent flow process. Fig. 6 presents such exemplary visualizations of superposing two and all transformations simultaneously. In each case, the latent evolution corresponds to simultaneous smooth variations of multiple image attributes. This indicates that our method also generalizes well to superposing different transformations.

Notice that we only apply single and separate transformations in the training stage. Switching or superposing transformations in the test phase can be thus understood as an extrapolation test to measure the generalization ability of the learned equivariance to novel compositions.

**Equivariance generalization to new data.** We also test whether the learned equivariance holds for Out-of-Distribution (OoD) data. To verify this, we validate our method on a test dataset that is different from the training set and therefore unseen to the model. Fig. 7 displays the exemplary visualization results of the VAE trained on MNIST [54] but evaluated on dSprites [59]. Although the reconstruction quality is poor, the learned equivariance is still clearly effective as each transformation still operates as expected: scaling, rotation, and coloring transformations from top to bottom respectively.

### 4.4 Results on Complex Real-world and Large-scale Datasets

Table 3 and 4 compare the equivariance error of our methods and the representative baselines on Falcol3D and Isaac3D, respectively. Notice that the values are much larger than previous datasets due to the increased image resolution. Our method still outperforms other baselines by a large margin

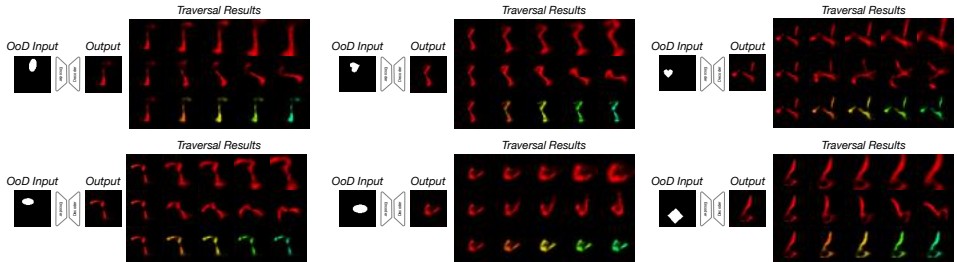

Figure 7: Equivariance generalization to unseen OoD input data. Here the model is trained on MNIST [54] but the latent flow is tested on dSprites [59].

| Methods | Lighting Intensity | Lighting X-dir | Lighting Y-dir | Lighting Z-dir | Camera X-pos | Camera Y-pos | Camera Y-pos |
|---|---|---|---|---|---|---|---|
| **TVAE** [45] | 11477.81 | 12568.32 | 11807.34 | 11829.33 | 11539.69 | 11736.78 | 11951.45 |
| **PoFlow** [79] | 8312.97 | 7956.18 | 8519.39 | 8871.62 | 8116.82 | 8534.91 | 8994.63 |
| **Ours** | **5798.42** | **6145.09** | **6334.87** | **6782.84** | **6312.95** | **6513.68** | **6614.27** |

Table 3: Equivariance error (↓) on Falcol3D [61].

and achieves reasonable equivariance error. Fig. 8 displays the qualitative comparisons of our method against other baselines. Our method precisely can control the image transformations through our latent flows. *Overall, the above results demonstrate that our method can go beyond the toy setting and can be further applied to more complex real-world scenarios.*

More visualization results of exemplary latent flows are kindly referred to in the supplementary.

## 5  Related work

**Disentangled representation learning.** The idea of learning disentangled representation dates back to factorizing non-redundant input patterns [74] but is recently first studied by InfoGAN [13] and $\beta$-VAE [35]. InfoGAN [13] achieves disentanglement by maximizing the mutual information between a subset of latent dimensions and observations, while $\beta$-VAE [35] induces the factorized posterior $q(\mathbf{z})$ by penalizing the Total Correlation (TC) through an extra hyper-parameter $\beta>1$ controlling the strength of the KL divergence. Following infoGAN, many attempts have been made to facilitate the discovery of semantically meaningful traversal directions through regularization [33, 42, 89, 34, 100, 66, 77, 90, 98, 84, 99, 78, 62]. The follow-up research of $\beta$-VAE mainly explored different methods to factorize the aggregated posterior [22, 25, 52, 46, 12, 44, 96, 23, 76, 58, 80, 28]. More recently, some works proposed to discover meaningful directions of diffusion models in the bottleneck of denoising networks [53, 64, 95, 41]. The previous literature mainly considers disentanglement as learning different transformations per dimension or per linear direction. Our method generalizes this concept to learning a distinct tangent bundle $\nabla u^k$ that moves every latent sample via dynamic OT.

We see the most similar method to ours is the work of [79]. In [79], the authors also apply the gradient of a potential function to move the latent code. However, their potentials are restricted to obey the wave equations, which do not really correspond to the OT theory. Also, they do not consider the posterior evolution but instead use the loss $||\mathbf{z}_t - \texttt{Encode}(\mathbf{x}_t)||^2$ to match the latent codes. By contrast, we propose a unified probabilistic generative model that encompasses the posterior flow that follows dynamic OT, the flow-like time evolution, and different supervision settings.

**Equivariant neural networks.** A function is said to be an equivariant map if it commutes with a given transformation, *i.e.,* $T'[f(x)] = f(T[x])$ where $T$ and $T'$ represent operators in different domains. Equivariance has been considered a desired inductive bias for deep neural networks as this property can preserve geometric symmetries of the input space [38, 75, 56, 57, 1]. Analytically equivariant

| Methods | Robot X-move | Robot Y-move | Camera Height | Object Scale | Lighting Intensity | Lighting Y-dir | Object Color | Wall Color |
|---|---|---|---|---|---|---|---|---|
| **TVAE** [45] | 8441.65 | 8348.23 | 8495.31 | 8251.34 | 8291.70 | 8741.07 | 8456.78 | 8512.09 |
| **PoFlow** [79] | 6572.19 | 6489.35 | 6319.82 | 6188.59 | 6517.40 | 6712.06 | 7056.98 | 6343.76 |
| **Ours** | **3659.72** | **3993.33** | **4170.27** | **4359.78** | **4225.34** | **4019.84** | **5514.97** | **3876.01** |

Table 4: Equivariance error (↓) on Isaac3D [61].

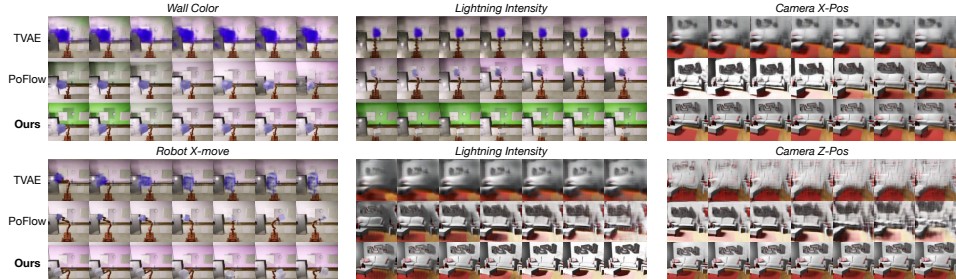

Figure 8: Qualitative comparison of our method against TVAE and PoFlow on Falcol3D and Isaac3D.

networks typically enforce explicit symmetry to group transformations in neural networks [16, 17, 68, 93, 92, 85, 31, 39]. Another line of research proposed to directly learn approximate equivariance from data [21, 18, 49, 20, 45]. Our framework re-defines approximate equivariance by matching the latent probabilistic flow to the actual path of the given transformation in the image space.

**Optimal transport in deep learning.** There is a vast literature on OT theory and applications in various fields [87, 88]. Here we mainly highlight the relevant applications in deep learning. The pioneering work of [19] proposed a light-speed implementation of the Sinkhorn algorithm for fast computation of entropy-regularized Wasserstein distances, which opened the way for many differentiable Sinkhorn algorithm-based applications [32, 29, 14, 27, 51]. In generative modeling, the Wasserstein distance is often used to minimize the discrepancy between the data distribution and the model distribution [2, 81, 72, 65]. Inspired by the fluid mechanical interpretation of OT [4], some normalizing flow methods [69, 24, 48] considered regularizing the velocity fields to satisfy the HJ equation, thus matching the dynamic OT plan [94, 30, 83, 63, 60]. Our method applies PINNs [67] to directly model generalized HJ equations in the latent space and uses the gradient fields of learned potentials to generate latent flows, which also aligns to the theory of dynamic fluid mechanical OT.

## 6    Conclusion

In this paper, we introduce Flow Factorized Representation Learning which defines a set of latent flow paths that correspond to sequences of different input transformations. The latent evolution is generated by the gradient flow of learned potentials following dynamic optimal transport. Our setup re-interprets the concepts of both *disentanglement* and *equivariance*. Extensive experiments demonstrate that our model achieves higher likelihoods on standard representation learning benchmarks while simultaneously achieving smaller equivariance error. Furthermore, we show that the learned latent transformations generalize well, allowing for flexible composition and extrapolation to new data.

## 7    Limitations

For flexibility and efficiency, we use PINN [67] constraints to model the HJ equation. However, such PDE constraints are approximate and not strictly enforced. Other PDE modeling approaches include accurate neural PDE solvers [40, 8, 70] or other improved PINN variants such as competitive PINNs [97] and robust PINNs [3]. Also, when infering with observed $k$, we change the posterior from $q(\bar{z}|\bar{x}, k)$ to $q(\bar{z}|x_0, k)$ because we assume $k$ contains sufficient information of the whole sequence. To keep the posterior definition of $q(\bar{z}|\bar{x}, k)$, we need to make $q(z_t)$ also a function of $x_t$. This can be achieved either by changing the potential to $u(z_{t-1}, x_t, t-1)$ or modifying the external driving force to $f(z_{t-1}, x_t, t-1)$. Nonetheless, we see these modifications would make the model less flexible than our current formulations as the element $x_t$ might be needed during inference.

## Acknowledgments and Disclosure of Funding

This work was supported by the MUR PNRR project FAIR (PE00000013) funded by the NextGeneration-EU, by the PRIN project CREATIVE (Prot. 2020ZSL9F9), by the EU H2020 project AI4Media (No. 951911), and by the Bosch Center for Artificial Intelligence.

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

# A  Supplementary Material

## A.1  Pseudo codes

```python
import torch

#Randomly sample a transformation at each iteration
index = torch.randint(0, potential_number)
x_bar = sequence_generation(index)

#Generating index according to the supervision setting
if training_mode = "supervised":
  index_potential = index
elif training_mode = "weakly-supervised":
  index_potential = q_k(x_bar)

#initial element of the sequence
z, rho_z = flow_vae(x_bar[0])

#Future elements of the sequence obtained by latent flow
for t in range(0,T)
    PDE_loss, delta_z, delat_rho_z = HJ_PDE(index_potential,z,t)

    #Updates in the sample and probability space
    z = z + delta_z
    rho_z = rho_z + delat_rho_z

    #Inference at every intermediate step
    hat_xt = flow_vae.inference(z)

    #Loss: PDE loss + reconstrutction loss + KL div
    loss += PDE_loss + CE(hat_xt,x_bar[t]) + KL(rho_z, prior_rho_z)

#KL div for index prediction (weakly-supervised setting)
if training_mode = "weakly-supervised":
    loss += KL(index_potential,index)

loss.backward()
optimizer.step()
```

Figure 9: Pytorch-like pseudo codes for training our flow-factorized VAE.

Fig. 9 displays the Pytorch implementation for training our flow-factorized VAE under different supervision settings. Here we omit the computation of HJ PDEs for concisity.

## A.2  Implementation details

**Common settings.** During the training stage, we randomly sample one single transformation at each iteration. The batch size is set to $128$ for both datasets. We use Adam optimizer and the learning rate is set as $1e{-}4$ for all the parameters. The encoder consists of four stacked convolution layers with the activation function ReLU, while the decoder is comprised of four stacked transposed convolution layers. For the prior evolution, the diffusion coefficient $D_k$ is initialized with $0$ and we set it as a learnable parameter for distinct $k$. For MLPs that parameterize the potential $u(z, t)$ and the force $f(z, t)$, we use the sinusoidal positional embeddings [86] to embed the timestep $t$, and use linear layers for embedding the latent code $z$. Tanh gates are applied as the activation functions of the MLPs. All the experiments are run on a single NVIDIA Quadro RTX 6000 GPU.

**MNIST.** The input images are of the size $28{\times}28$. The sequence of each transformation contains 9 states of variations. The scaling transformation scales the image from $1.0$ up to $1.8$ times. The rotation transformation rotates the object by maximally $80$ degrees, and the coloring transformation adjusts the image hue from $0$ to $340$ degrees. The model is trained for $90,000$ iterations.

**Shapes3D.** The input images are resized to $64\times64$. Each transformation sequence consists of $8$ images. The model is also trained for $90,000$ iterations.

**Falcol3D and Isaac3D.** The input images are in the resolution of $128\times128$. We use the self-contained transformations of the datasets, which mainly comprise variations of lighting conditions and viewpoints in indoor 3D scenes for Falcolr3D, and different robot arm movements in dynamic environments for Isaac3D.

**Weakly-supervised setting.** For the `Gumbel-Softmax` trick, we re-parameterize $q_\gamma(k|\bar{x})$ by

$$y_i = \frac{e^{\frac{x_i+g_i}{\tau}}}{\sum_i e^{\frac{x_i+g_i}{\tau}}} \tag{15}$$

where $x_i$ is the category prediction, $g_i$ is the sample drawn from Gumbel distributions, and $\tau$ is the small temperature to make `softmax` behave like `argmax`. We take the 'hard' binary prediction in the forward pass and use the straight-through gradient estimator [6] during backpropagation. The temperature $\tau$ is initialized with $1$ and is gradually reduced to $0.05$ with the annealing rate $3e-5$.

**Baselines.** For the disentanglement methods, we largely enrich the original MNIST dataset by adding the transformed images of the whole sequence. This makes it possible for both $\beta$-VAE and FactorVAE to learn the given transformations in an unsupervised manner. For tuning the interpolation range, we start from the initial value $z_i$ and traverse till the appropriate bound which is selected from the range $[-5, 5]$ with the interval of $0.1$.

### A.3 Disentanglement metrics

There are many traditional disentanglement metrics [71, 26, 12], but they are designed for single-dimension traversal methods. These metrics assume and require that each latent dimension is responsible for one semantic and manipulating single dimensions of the latent variable would involve distinct output transformations. However, for the recent disentanglement methods including ours [77, 84, 79], there emerges a more realistic disengagement setting: all the latent dimensions are perturbed by vectors for meaningful output variations. When it comes to these vector-based disentanglement methods, their scores of disentanglement metrics would drop considerably and cannot be compared with those single-dimension baselines.

<table>
<tr><td colspan="6" align="center">Table 5: VP Scores (%) on MNIST.</td></tr>
<tr><td>Training Set Split</td><td>Ours</td><td>PoFlow</td><td>TVAE</td><td>FactorVAE</td><td>$\beta$-VAE</td></tr>
<tr><td>10%</td><td>**95.69**</td><td>93.05</td><td>89.91</td><td>85.92</td><td>87.31</td></tr>
<tr><td>1%</td><td>**92.71**</td><td>91.27</td><td>88.15</td><td>84.46</td><td>85.25</td></tr>
</table>

<table>
<tr><td colspan="6" align="center">Table 6: VP Scores (%) on Shapes3D.</td></tr>
<tr><td>Training Set Split</td><td>Ours</td><td>PoFlow</td><td>TVAE</td><td>FactorVAE</td><td>$\beta$-VAE</td></tr>
<tr><td>10%</td><td>**95.92**</td><td>91.48</td><td>88.27</td><td>84.49</td><td>85.91</td></tr>
<tr><td>1%</td><td>**77.03**</td><td>72.32</td><td>68.39</td><td>63.83</td><td>65.78</td></tr>
</table>

Nonetheless, certain disentanglement metrics such as VP scores [100] can be leveraged as they do not pose any assumptions on the latent space but only require image pairs $[x_0, x_T]$ of different transformations for evaluation. The VP metric adopts the few-shot learning setting (using $1\%$ or $10\%$ of the dataset as the training set) and takes a lightweight neural network for learning to classify image pairs $[x_0, x_T]$ of different attributes. The generalization ability ( *i.e.,* validation accuracy) can be thus regarded as a reasonable surrogate for the disentanglement ability. Table 5 and 6 present the VP scores of all the baseline methods on MNIST and Shapes3D. To ensure a fair comparison, for FactorVAE and $\beta$-VAE, we choose the dimensions with the lowest equivariance errors to generate image pairs of different transformations. Our method outperforms the previous disentanglement baselines and achieves superior performance on the VP scores. This indicates that our flow-factorized VAE has better disentanglement ability.

### A.4 Ablation studies

**Impact of different priors.** We use diffusion equations to model the prior evolution as random particle movement. It would also be intriguing to choose other priors commonly used in the VAE literature, such as Standard Gaussian (SG) priors $\mathcal{N}(0, 1)$, mixture of Gaussian (MoG) priors $\sum w_i \mathcal{N}(\mu_i, \sigma_i^2)$, and VAMP priors [82] which average aggregated posterior of $N$ pseudo-inputs as $1/N \sum_n q(z_n)$. Table 7 presents the equivariance error of different priors on MNIST. Among these priors, our diffusion equations achieve the best performance. This meets our assumption that modeling the prior evolution

Table 7: Equivariance error of different priors.

| Prior | Scaling | Rotation | Coloring |
|---|---|---|---|
| SG | 190.24±2.18 | 158.93±3.25 | 164.18±2.77 |
| MoG | 188.23±2.45 | 157.79±2.86 | 161.49±2.62 |
| VAMP | 192.81±3.67 | 161.47±4.12 | 162.97±3.89 |
| **Diffusion** | **185.42±2.35** | **153.54±3.10** | **158.57±2.95** |

Table 8: Equivariance error of different PDEs.

| Prior | Scaling | Rotation | Coloring |
|---|---|---|---|
| Heat | 223.95±3.38 | 212.47±3.85 | 207.66±2.91 |
| FP | 211.54±3.17 | 188.59±3.92 | 194.73±3.09 |
| OHJ | 190.43±2.48 | 163.87±3.03 | 162.38±2.86 |
| **GHJ** | **185.42±2.35** | **153.54±3.10** | **158.57±2.95** |

as a diffusion process suits more the random motion. Nonetheless, we see that the performance gap between each baseline is narrow, which somehow implies that the impact of different priors is limited.

**Impact of different PDEs.** We apply the generalized HJ (GHJ) equation as the PINN constraint in order to achieve dynamic OT. It would be also interesting to try other commonly used PDEs. We compare our GHJ with the ordinary HJ (OHJ) equation, the Fokker Planck (FP) equation, and the heat equation. Table 8 compares the equivariance error of PDEs on MNIST. Our GHJ and OHJ equations achieve the best performance as they both satisfy the condition of dynamic OT. This empirical evidence indicates that the OT theory can indeed model better latent flow paths. Moreover, our GHJ outperforms the OHJ by a slight margin. We attribute this advantage to the external driving force $f(z, t)$ which gives us more flexibility and dynamics in modeling the velocity fields $\nabla u^k$.

Table 9: Equivariance error on MNIST of a different number of transformations ($K$).

| $K$ | Scaling | Rotation | Coloring |
|---|---|---|---|
| 1 | 185.27±2.59 | – | – |
| 2 | 185.78±2.21 | 154.29±2.87 | – |
| 3 | 185.42±2.45 | 153.54±3.10 | 158.57±2.95 |

Table 10: Equivariance error on MNIST of different sequence lengths ($T$).

| Sequnce Length ($T$) | Scaling | Rotation |
|---|---|---|
| 9 | 185.42±2.35 | 153.54±3.10 |
| 12 | 214.47±2.59 | 198.72±2.89 |

**Impact of different $K$.** We conduct an ablation study on the impact of the number of transformations on MNIST and present the evaluation results in Table 9. As indicated above, in general, the performance is not affected by the number of transformations being applied. The fluctuation of the results when $K$ varies can be sufficiently negligible. We expect that this is because the transformations are learned by distinct potentials (which are implemented as $K$ different MLPs). Each flow evolves along with the gradient field $\nabla u$ on the potential landscape $u$; *having multiple latent flows defined on different potential landscapes therefore does not interfere with each other.*

**Impact of different sequence lengths.** Regarding the impact of sequence lengths, if the sequence is longer and has larger variations, generally the equivariance error would be worse. To better illustrate this point, we perform an ablation study on MNIST and present the results in Table 10. Specifically, we change the sequence length from 9 to 12, which increases the extent of scaling from maximally 1.8 times to maximally 2.1 times, and increases the rotation angle from maximally 80 degrees to maximally 110 degrees. As can be observed, the equivariance error gets larger when the sequence becomes longer and the variations are larger. Notice that even for the longer sequence, our method still outperforms other baselines with shorter sequences.

## A.5 HJ equations as dynamic optimal transport

We now turn to introduce why HJ equations could minimize the Wasserstein distance. As stated in [4], the $L_2$ Wasserstein distance can be re-formulated in the fluid mechanical interpretation as

$$W^2 = \inf \int_D \int_0^1 \frac{1}{2} \rho(x, t) v(x, t)^2 \, dx \, dt \tag{16}$$

where the density satisfies the continuity equation ($\partial_t \rho = -\nabla \cdot (\rho(x, t) v(x, t))$. If we introduce the momentum $m(x, t) = \rho(x, t) v(x, t)$ and two Lagrange multipliers $u$ and $\lambda$, the Lagrangian function of the Wasserstein distance would be:

$$L(\rho, m, \phi) = \int_D \int_0^1 \frac{||m||^2}{2\rho} + u(\partial_t \rho + \nabla \cdot m) - \lambda(\rho - s^2) \tag{17}$$

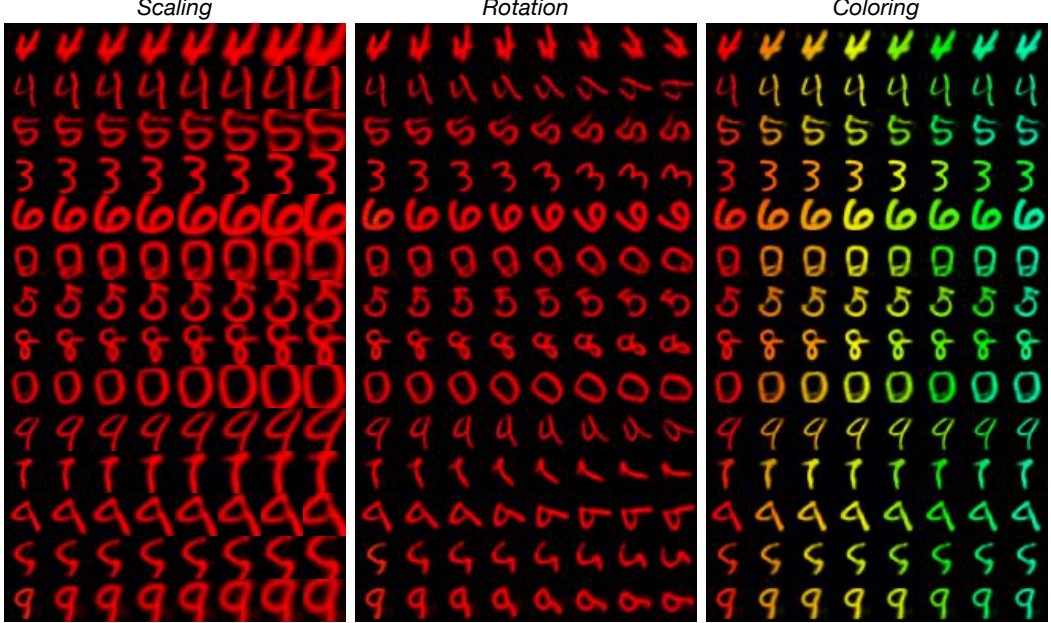

Figure 10: More visualizations of the learned latent flows on MNIST [54].

where the second term is the equality constraint, and the third term is an equality constraint with a slack variable $s$. Using integration by parts formula, the above equation can be re-written as

$$L(\rho, m, \phi) = \int_D \int_0^1 \frac{||m||^2}{2\rho} + \int_D u\rho|_0^1 - \int_D \int_0^1 (\partial_t u\rho + \nabla u \cdot m) - \lambda(\rho - s^2) \qquad (18)$$

Based on the set of Karush–Kuhn–Tucker (KKT) conditions ($\partial_m L = 0$, $\partial_u L = 0$, $\partial_\rho L = 0$, and $\lambda \geq 0$), we would have:

$$\begin{cases} \partial_m L = \frac{m}{\rho} - \nabla u = v - \nabla u = 0 \\ \partial_u L = \partial_t \rho + \nabla \cdot m = 0 \\ \partial_\rho L = -\frac{||m||^2}{2\rho^2} - \partial_t u - \lambda = -\frac{1}{2}||v||^2 - \partial_t u - \lambda = 0 \end{cases} \qquad (19)$$

where the first condition indicates that the gradient $\nabla u$ acts as the velocity field, and the third condition implies the optimal solution is given by the generalized HJ equation:

$$\partial_t u + \frac{1}{2}||\nabla u||^2 = -\lambda \leq 0 \qquad (20)$$

We thus apply the generalized HJ equation (*i.e.*, $\partial_t u + \frac{1}{2}||\nabla u||^2 \leq 0$) as the constraints. We further use an extra negative force because this would give more dynamics for modeling the posterior flow.

### A.6 More visualizations

Fig. 10, 11, and 12 display more visualization results of the latent evolution on MNIST, Shapes3D, Falcol3D and Isaac3D, respectively. Across all the datasets, our method presents precise control of the given transformations. Fig. 13 and 14 show more latent evolution results of switching transformations (top) and combining transformations (bottom) on MNIST and Shapes3D, respectively. Fig. 15 also visualizes a few examples of superposing and switching transformation on Falcol3D and Isaac3D. Our latent flows learn to compose or switch different transformations precisely and flexibly.

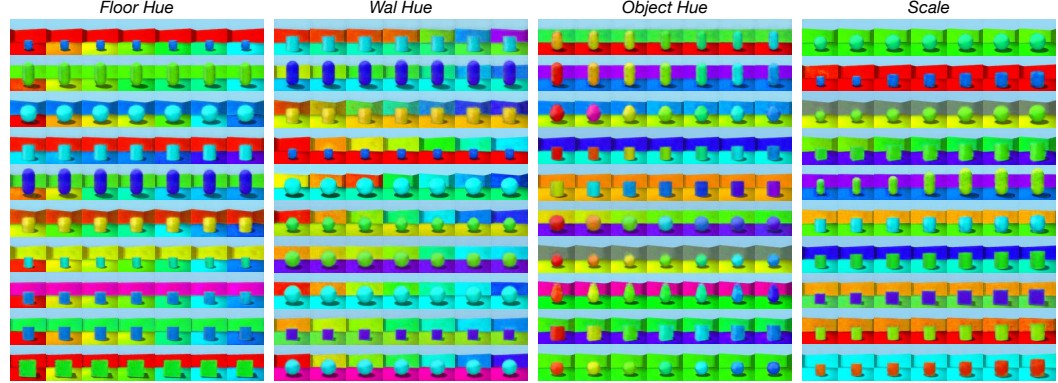

Figure 11: More visualizations of the learned latent flows on Shapes3D [10].

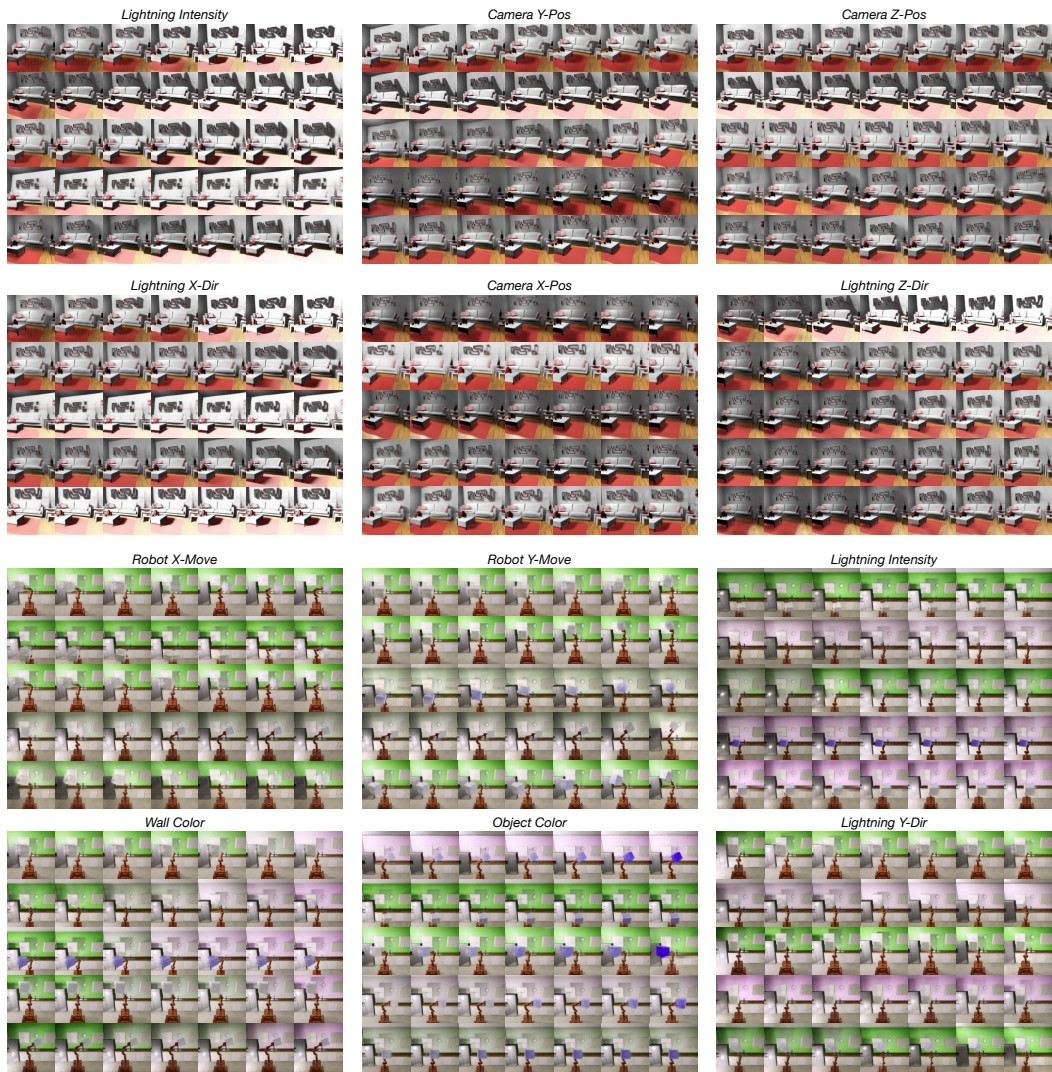

Figure 12: Exemplary visualizations of learned latent flows on Falcol3D (*top*) and Isaac3D (*bottom*).

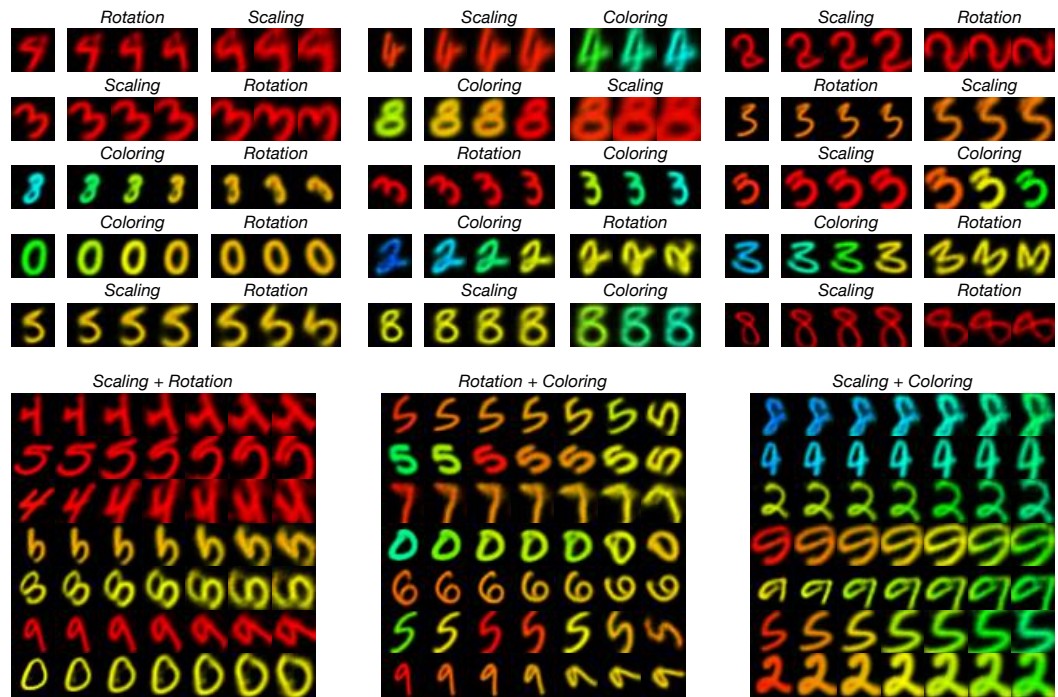

Figure 13: More visualizations of switching and superposing transformations on MNIST [54].

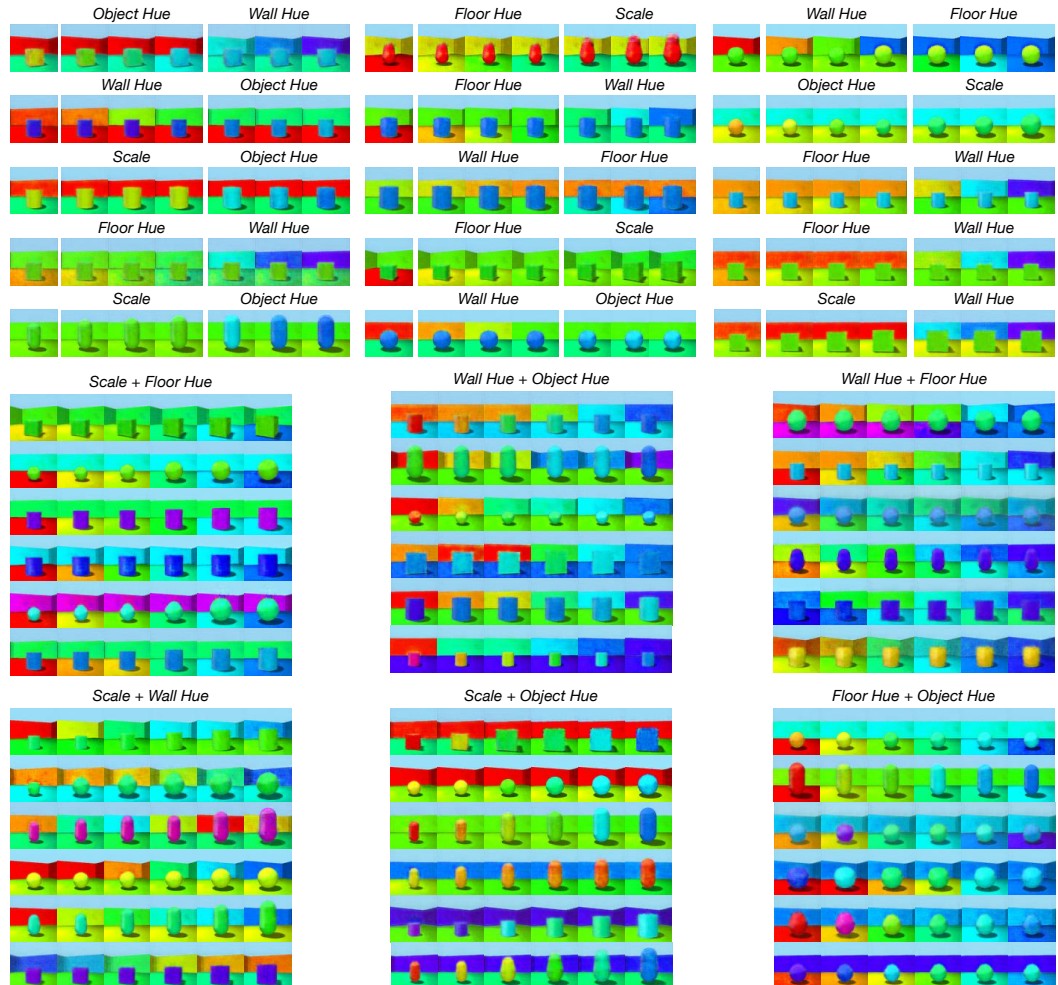

Figure 14: More visualizations of switching and superposing transformations on Shapes3D [10].

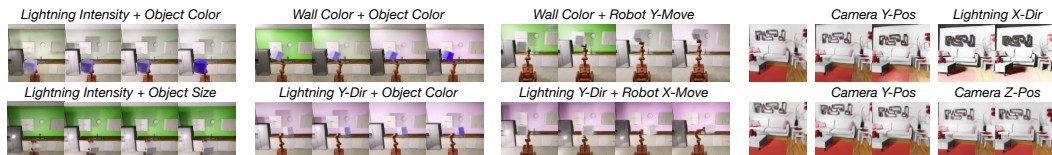

Figure 15: Exemplary visualization results of superposing transformations on Isaac3D (*left*) and switching transformations on Falcol3D (*right*).

