# OpenReview forum: "Flow Factorized Representation Learning"
_NeurIPS.cc/2023/Conference — NeurIPS 2023 poster_

### Official Review · Reviewer_wDok · 2023-07-05

**Soundness:** 3 good
**Presentation:** 3 good
**Contribution:** 3 good
**Rating:** 6
**Confidence:** 3

**Summary:**

The paper introduces a new generative model called Flow Factorized Representation Learning that learns a distribution over sequences of observed variables using which one can learn both disentangled and equivariant representations. The model is based on the idea of representing transformations in the input space as latent flow paths generated by the gradient fields of learned potentials. In this way, disentanglement can be read from the set of tangent directions, while equivariance can be read by analyzing whether the probabilistic path in the input space and the corresponding latent path result in the same latent representation. The paper includes detailed formal derivation of the model which is realized as a Variational Autoencoder. Experiments are performed on MNIST and 3DShapes datasets.


**Strengths:**

- Solid model derivation
- Strong experimental results showing empirical equivariance and disentanglement, though on basic datasets.


**Weaknesses:**

W1) It is not easy to follow the derivations, and it is not clear how the training is performed at the end. It would be helpful to at least add a pseudocode of the algorithm.

W2) It is not clear to me what exactly are the input sequences of observations in the experimental section. In other words, how big is T and how to you obtain the intermediate input datapoints between x0 and xT?

W3) It is unclear what is the complexity of the model.

W4) Experiments are performed on very simple datasets. It is even argued in line 33 that prior methods have not gone beyond toy settings but this work does not do that either.


**Questions:**

1. What is the sequence length and how does this affect the results? Could you please add more details on how you obtain the input sequences?
2. How would this method generalize to more complex data? While the method demonstrates strong results on MNIST and 3DShapes, these datasets are rather basic.
3. In the Extrapolation case, how do you aggregate representations from each of the flows?


**Limitations:**

- Limitations are discussed.
- No code provided.

---

> ### Author Rebuttal · Authors · 2023-08-09
>
> We thank $\textcolor{pink}{Reviewer\ wDok}$ for the positive feedback and constructive suggestions. Below we respond to the concerns point by point.
>
> ---
>
> **Q1. Pseudocode of the Algorithm**
>
> Thanks for the constructive suggestion. We provide the following pseudo-codes for training our flow-factorized VAE:
>
> ```python
> import torch
>
> #Randomly sample a transformation at each iteration
> index = torch.randint(0, potential_number)
> x_bar = sequence_generation(index)
>
> #Generating index according to the supervision setting
> if training_mode = "supervised":
>   index_potential = index
> elif training_mode = "weakly-supervised":
>   index_potential = q_k(x_bar)
>
> #initial element of the sequence
> z, rho_z = flow_vae(x_bar[0])
>
> #Future elements of the sequence obtained by latent evolution
> for t in range(0,T)
>    PDE_loss, delta_z, delat_rho_z = HJ_PDE(index_potential,z,t)
>
>    #Updates in the sample and probability space
>    z = z + delta_z
>    rho_z = rho_z + delat_rho_z
>
>    #Inference at every intermediate steps
>    hat_xt = flow_vae.inference(z)
>
>    #Loss: PDE loss + reconstrutction loss + KL div
>    loss += PDE_loss + CE(hat_xt,x_bar[t]) + KL(rho_z, prior_rho_z)
>
> #KL div for index prediction (weakly-supervised setting)
> if training_mode = "weakly-supervised":
>    loss += KL(index_potential,index)
>
>
> loss.backward()
> optimizer.step()
> ```
>
> Here we omit the computation of HJ PDEs for concisity. We will add the above pseudo-codes and an algorithm to the revised supplementary material.
>
> **Q2. Sequence Length and Intermediate Transformations**
>
> As stated in line13 and line16 of the supplementary, the sequence length (T) for MNIST and Shapes3D is 9 and 8, respectively. For MNIST, we transform the data using the image geometric transformation functions imported from the $\texttt{Kornia}$ library. For Shapes3D, this dataset itself has sequences of different transformations so we just use its self-contained sequences.
>
> As for the impact of sequence length, if the sequence is longer and has more variations, generally the equivariance error would be worse. To better illustrate this point, we did an ablation study on MNIST. Specifically, we change the sequence length from 9 to 12, which increases the extent of scaling from maximally 1.8 times to maximally 2.1 times, and increases the rotation angle from maximally 80 degrees to maximally 110 degrees. The Table below presents the results.
>
> Equvariance error on MNIST for different sequence lengths
> Sequence Length|Scaling|Rotation|
> |:---:|:---:|:---:|
> |9|185.42±2.35|153.54±3.10|
> |12|214.47±2.59|198.72±2.89|
>
> As can be seen above, the equivariance error is larger as the sequence is longer and the variations are larger. Notice that even for the longer sequence, our method still outperforms other baselines with shorter sequences.
>
> **Q3. Computational Complexity**
>
> We present the training time cost of our method on MNIST in the Table below, as well as the results of TVAE and PoFlow. In the supervised setting, our method is slightly more time-consuming than PoFlow (around 10%) and than TVAE (around 20%). When it comes to the weakly-supervised fashion, our flow-factorized VAE costs more time, as we need to use the Gumbel-Softmax trick for predicting the transformation index given the whole image sequence ($q(k,\bar{\mathbf{x}})$) and aggregate flows according to the index prediction ($\mathbf{z} _{t+1} = \mathbf{z} _t + \sum _i^K q(k,\bar{\mathbf{x}}) _i  \nabla u^i$). Nonetheless, we think the overall time cost is still acceptable considering the performance gain over the baselines (especially for the supervised setting).
>
>    Training time cost on MNIST every iteration
> ||Ours (weakly-supervised)|Ours (supervised)|PoFlow|TVAE|
> |:---:|:---:|:---:|:---:| :---:|
> |Time (s)|1.076|0.723|0.638|0.591|
>
>
> **Q4. Application on Complex Real-world Datasets**
>
> Thanks for the constructive advice! We follow your suggestion and take a step further to evaluate our method on Falcor3D and Isaac3D, two complex large-scale and real-world datasets that contain sequences of different transformations. Falcol3D consists of indoor 3D scenes in different lighting conditions and viewpoints, while Isaac3D is a dataset of various robot arm movements in dynamic environments.
>
> Due to the limited space, please refer to the response to $\textcolor{red}{Reviewer\ tD2D}$ and the attached one-page pdf for the relevant results and discussions.
>
> **Q5. Flow aggregation in extrapolation**
>
> As indicated in line217 of the paper, the aggregation is simply done by linearly superposing different gradient fields. For example, if we would like to superpose scaling ($u^1$) and rotation ($u^2$), the new latent evolution would follow:
>
> $\mathbf{z} _t = \mathbf{z} _{t-1} + (\nabla u^1 + \nabla u^2)$
>
>
> **Q6. Code would be released upon acceptance**
>
> Thanks for your interest! Our code is implemented in Pytorch and will be publicly released upon acceptance!
>
> ---
>
> Thanks again and it is more than welcome to post further comments.

---

> > ### Comment · Reviewer_wDok · 2023-08-14
> > **Thanks for the comments**
> >
> > Thanks for providing the thorough rebuttal. You addressed all my questions and I particularly appreciate the extra experiments on the two more complex dataset. I would recommend to add these to the main part of the paper if possible as they nicely highlight the strengths of the method.

---

> > > ### Author Response · Authors · 2023-08-14
> > > **Thanks for the encouraging feedbak; We would definitely include extra experiments.**
> > >
> > > Thanks for the encouraging feedback!
> > >
> > > We would definitely include the results and analysis of the extra experiments in the revised paper.
> > >
> > > We sincerely appreciate your thoughtful review and valuable comments, as well as the time you spent on our paper!

---

### Official Review · Reviewer_znKv · 2023-07-06

**Soundness:** 2 fair
**Presentation:** 3 good
**Contribution:** 2 fair
**Rating:** 5
**Confidence:** 3

**Summary:**

This paper proposes a flow factorized representation learning approach for disentanglement and equivariant representation learning in a learned latent space. This work introduces a generative model that defines a set of latent probability paths to specify different input transformations and each latent flow is generated following dynamic optimal transport. Results show that the proposed framework is able to learn both more efficient and more usefully structured representations and thus achieve improved performance on representation learning benchmarks.

**Strengths:**

1. This work provides a new understanding of the disentanglement and equivariant, which are important problems for representation learning research.
2. This paper proposes flow factorized representation learning which is a new and reasonable viewpoint for structured representation learning.
3. Visualization results demonstrate that the proposed framework is able to switch transformations during the latent sample evolution and achieves both disentanglement and equivariance.
4. Experimental results show that the learned latent transformations have flexibility and can generalize to new data.

**Weaknesses:**

1. The datasets used in this work are toy (e.g., MNIST, Shapes3D). Large-scale and real-world datasets are expected to evaluate the efficacy of the proposed approach, such as CIFAR, ImageNet.
2. Advanced baseline methods are expected to be compared including StyleGAN, StyleGAN2, and Diffusion models.
3. Can the learned disentangled representations be transferred across datasets, such as from Shapes3D to MNIST?

**Questions:**

Please refer to detailed comments in weaknesses.

**Limitations:**

This works shows no potential negative societal impact.

---

> ### Author Rebuttal · Authors · 2023-08-09
>
> We thank $\textcolor{green}{Reviewer\ znKv}$ for the constructive suggestions. Below we respond to the concerns point by point.
>
> ---
>
> **Q1. Application on Real-world Datasets**
>
> Thanks for the great advice! As suggested by $\textcolor{magenta}{Reviewer\ ww47}$, we take a step further to apply our method on Falcol3D and Isaac3D, two complex large-scale and real-world datasets that contain sequences of different transformations. Falcol3D consists of indoor 3D scenes in different lighting conditions and viewpoints, while Isaac3D is a dataset of various robot arm movements in dynamic environments.
>
> Equivariance errors on Falcor3D
> |Methods|Lighting Intensity|Lighting X-dir|Lighting Y-dir|Lighting Z-dir|Camera X-pos|Camera Y-pos|Camera Z-pos|
> |:---:|:---:|:---:|:---:| :---:| :---:| :---:|  :---:|
> |TVAE|11477.81|12568.32|11807.34|11829.33|11539.69|11736.78|11951.45|
> |PoFlow|8312.97|7956.18|8519.39|8871.62|8116.82|8534.91|8994.63|
> |Ours|**5798.42**|**6145.09**|**6334.87**|**6782.84**|**6312.95**|**6513.68**|**6614.27**|
>
> Equivariance errors on Issac3D
> |Methods|Robot X-move|Robot Y-move|Camera Height|Object Scale|Lighting Intensity|Lighting Y-dir|Object Color|Wall Color|
> |:---:|:---:|:---:|:---:| :---:| :---:| :---:|  :---:| :---:|
> |TVAE|8441.65|8348.23|8495.31|8251.34|8291.70|8741.07|8456.78|8512.09|
> |PoFlow|6572.19|6489.35|6319.82|6188.59|6517.40|6712.06|7056.98|6343.76|
> |Ours|**3659.72**|**3993.33**|**4170.27**|**4359.78**|**4225.34**|**4019.84**|**5514.97**|**3876.01**|
>
>
>
> The two Tables above present the equivariance errors of our method and TVAE/PoFlow. Notice that the values are much larger due to the increased resolution. Our method still outperforms other baselines by a large margin and achieves reasonable equivariance error. To visually display the learned latent transformations, we also conducted some qualitative evaluations and put them in the attached one-page pdf (which comprises exemplary latent evolutions of different transformations and exemplary comparisons against baselines). Our method can precisely control the image transformations through our latent flows.
>
>
>
>
> **Q2. Comparison against StyleGANs and Diffusion Models**
>
> It is unlikely to compare our method with StyleGANs or diffusion models because their generative models are fundamentally different from ours. Our flow factorized VAE is defined on **sequences of data transformations**, while StyleGANs and diffusion models, despite being super powerful in generative modeling, are in essence generative models defined on **single images**. _Moreover, StyleGANs and diffusion models possess inherent architectural limitations of being simultaneously equivariant and disentangled._
>
> StyleGAN models and GANs in general do not have any encoders which could define the input space; the input to GANs is basically random noises. This architectural limitation makes it almost impossible to have equivariant image transformations. There exists a research line called GAN inversion which could identify the latent codes which closely reconstruct given images [a]. With the technique of GAN inversion, it might be possible to have an approximate equivariance mapping. However, the GAN inversion techniques usually require optimization for every single image, which makes it very complex and challenging to incorporate GANs into our framework.
>
> For diffusion models, there is no concept of a well-defined latent space as they generate images by recursive denoising from a Gaussian. Consequently, there does not exist such a definition or concept of disentanglement for diffusion models. Some recent works hypothesize that the bottleneck of the diffusion network might contain some semantics [b]. However, the semantics discovery in the bottleneck requires complex optimization processes and even needs the help of the CLIP text encoder, which again makes it really challenging to design special diffuson models for our framework.
>
> In a nutshell, the property of disentanglement and equivariance restricts us to implement our framework as encoder-decoder-like deep architectures for now. Nonetheless, we see that the flow-factorized VAE is a first step and we would keep investigating extensions to other generative models.
>
>
> > [a] PTI: Pivotal Tuning for Latent-based editing of Real Images. ACM TOG20.
> >
> > [b] Diffusion models already have a semantic latent space. ICLR23.
>
>
> **Q3. Transferability of Learned Transformations**
>
> The transferability depends on the specific experimental setting. If we would like to validate an already-trained model on a new dataset, the reconstruction quality might be poor but the latent flow operator should be transferred (as indicated in Fig. 7 of the paper).
>
> On the other hand, if we consider training an already-trained model on a new dataset, the learned transformations are not likely to be transferred because the model and the latent space would be completely different (as a consequence of training on new data). The corresponding latent flows would not lead to the prescribed transformations.
>
> ---
>
> Thanks again and it is more than welcome to ask further questions.

---

> > ### Comment · Reviewer_znKv · 2023-08-21
> > **Response to the author rebuttal**
> >
> > Thanks to the authors for the rebuttal. Most of my concerns have been addressed. I decided to raise my score.

---

> > > ### Author Response · Authors · 2023-08-21
> > > **Thanks for raising the score!**
> > >
> > > Thanks for raising the score and the positive feedback!
> > >
> > > We appreciate your constructive suggestions and valuable comments.

---

### Official Review · Reviewer_xnNs · 2023-07-06

**Soundness:** 3 good
**Presentation:** 3 good
**Contribution:** 3 good
**Rating:** 6
**Confidence:** 2

**Summary:**

This paper proposes the Flow Factorized Representation to learn structured representations. It introduces a generative model that defines different input transformations with a set of latent flow paths. The latent evolution is generated by the gradient flow of learned potentials, which are parameterized by a set of MLPs. A PINN loss is introduced to ensure that the latent posterior flow follows the OT path. The proposed method is tested on the MNIST and Shapes3D datasets.

**Strengths:**

- The paper is well-written, and the proposed method is technically sound.
- The learned transformation are well factorized and demonstrate flexible composability.

**Weaknesses:**

- The paper lacks a comparison in terms of disentanglement. Since the authors claim that the proposed method brings new understandings to both disentanglement and equivariance, it would be beneficial to compare it with state-of-the-art disentanglement methods.  As traditional disentanglement metrics in VAE are unsuitable for this method, the VP metric [a] and the correlation metric [b] could be helpful. Additionally, a qualitative comparison would also be beneficial.
- More state-of-the-art disentanglement methods should be included in the comparison, such as [c, d].


[a] Zhu, Xinqi, et al. "Learning disentangled representations with latent variation predictability." ECCV 2020.
[b] Shen, Yujun, et al. "Interfacegan: Interpreting the disentangled face representation learned by gans." TPAMI 2020.
[c] Estermann, Benjamin, et al. "DAVA: Disentangling Adversarial Variational Autoencoder." ICLR 2023.
[d] Shao, Huajie, et al. "Controlvae: Controllable variational autoencoder." ICML, 2020.

**Questions:**

- What value is $k$ set to in the experiments? If $k$ is larger than the actual factors of the datasets, how will the model perform?

**Limitations:**

yes

---

> ### Author Rebuttal · Authors · 2023-08-09
>
> We thank $\textcolor{blue}{Reviewer\ xnNs}$ for the careful review and valuable advice. Below we address the concerns in detail.
>
> ---
>
> **Q1. Disentanglement Metrics**
>
> Thanks for the constructive suggestions! We strongly agree that the VP metric [a] indeed suits our method as it does not pose any assumptions on the latent space but only requires image pairs $\mathbf{x} _0,\mathbf{x} _T$ of different transformations for evaluation. The VP metric adopts the few-shot learning setting (using $1$\% or $10$\% of the dataset as the training set) and takes a light-weighted neural network for learning to classify image pairs $\mathbf{x} _0,\mathbf{x} _T$ of different attributes. The generalization ability (validation accuracy) can be thus regarded as a reasonable surrogate for the disentanglement ability. The attribute correlation used in InterfaceGAN is also a good metric but it requires pre-trained attribute estimators to get scores, which is usually used for face image datasets.
>
>
> VP Scores (%) on MNIST
> |Training Set Split|Ours|PoFlow|TVAE|FactorVAE|$\beta$-VAE|
> |:---:|:---:|:---:|:---:| :---:| :---:|
> |10%|**95.69**|93.05|89.91|85.92|87.31|
> |1%|**92.71**|91.27|88.15|84.86|85.25|
>
>
> VP Scores (%) on Shapes3D
> |Training Set Split|Ours|PoFlow|TVAE|FactorVAE|$\beta$-VAE|
> |:---:|:---:|:---:|:---:| :---:| :---:|
> |10%|**95.92**|91.48|88.27|84.49|85.91|
> |1%|**77.03**|72.32|68.39|63.83|65.78|
>
>
> We adopt the VP metric and evaluate the performance of the baseline methods on MNIST and Shapes3D. To ensure a fair comparison, for FactorVAE and $\beta$-VAE, we choose the dimensions with the lowest equivariance errors to generate image pairs of different transformations. As indicated in the above two Tables, our method still achieves superior performance on the VP scores and outperforms the previous disentanglement baselines. This indicates that our flow-factorized VAE has better disentanglement performance.
>
>
> >[a] Learning disentangled representations with latent variation predictability. ECCV20.
>
>
> **Q2. Qualitative Comparison**
>
> Since the datasets used in the paper (MNIST and Shapes3D) are relatively simple, it is hard to visually inspect the difference unless cherry-picking extreme samples. However, as we have further applied our method to Falcol3D and Isaac3D (the two more complex real-world datasets), we did some qualitative comparisons on these two datasets. Please refer to the attached one-page pdf for the exemplary visual comparisons. Our method has obviously better equivariance properties and improves the image attribute control compared with other baselines.
>
>
>
> **Q3. Added Baselines**
>
> Thanks for sharing the interesting works! DAVA [b] proposes an adversarial training procedure of VAEs to avoid manually tuning the dataset-specific regularization strength, while ControlVAE [c] defines a controller to automatically tune the regularization strength based on the proportional-integral-derivative (PID) control.
>
> Since both DAVA and ControlVAE can be regarded as improved extensions of $\beta$-VAE, we use the same strategy of processing the results of $\beta$-VAE to search for the best latent dimension that corresponds to the given transformation. The Table below displays the equivariance errors and VP scores of these two baselines against our methods. We would add these papers to the revised references and meanwhile do a more thorough literature review.
>
>
> Equivariance error ($\downarrow$) and VP scores ($\uparrow$) of DAVA and ControlVAE on MNIST
> Methods|Scaling|Rotation|Coloring|VP(10%)|VP(1%)|
> |:---:|:---:|:---:|:---:|:---:|:---:|
> |DAVA| 547.93±4.32| 523.98±5.47| 493.49±4.91|89.32|87.94 |
> |ControlVAE| 586.29±4.16|546.72±5.93 | 501.92±5.12|88.45|86.82 |
> |Ours|**185.42±2.35**|**153.54±3.10**|**158.57±2.95**|**95.69**|**92.71**|
>
>
> >[b] DAVA: Disentangling Adversarial Variational Autoencoder. ICLR23.
> >
> >[c] Controlvae: Controllable variational autoencoder. ICML20.
>
>
> **Q4. Choosing the Value of $k$**
>
> For MNIST, we choose the value of $k$ in line with TVAE and PoFlow. For Shapes3D, we use the self-contained transformations of the dataset but omit the transformation of "Object Shape". This is because the sequence is very short (4 elements) and its transition is very abrupt, which does not define a smooth and natural transformation like the other sequences.
>
> Equivariance error of different k on MNIST
> k|Scaling|Rotation|Coloring|
> |:---:|:---:|:---:|:---:|
> |1|185.27±2.59|-|-|
> |2|185.78±2.21|154.29±2.87|-|
> |3|185.42±2.45|153.54±3.10|158.57±2.95|
>
> We did an ablation study on the impact of $k$ on MNIST and present the evaluation results in the above Table. As indicated above, in general, the performance is not affected by the number of transformations being applied. The fluctuation of the results when $k$ varies can be sufficiently negligible. We expect that this is because the transformations are learned by distinct potentials (which are implemented as k different MLPs). Each flow evolves along with the gradient field $\nabla u$ on the potential landscape $u$; _**having multiple latent flows defined on different potential landscapes therefore does not interfere with each other**_.
>
> Since the value of $k$ is defined exactly the same as the number of transformations being applied, it is unlikely to set a $k$ larger than the actual number of given sequences. However, it might be possible to consider a larger $k$ in some unsupervised methods such as [d,e]. Then the model is likely to learn various combinations of the actual variation factors of the dataset.
>
> > [d] Unsupervised discovery of interpretable directions in the gan latent space. ICML20.
> >
> > [e] Latent Traversals in Generative Models as Potential Flows. ICML23.
>
> ---
>
> Thanks again and it is more than welcome to post further comments.

---

> ### Comment · Reviewer_xnNs · 2023-08-18
> **Response to the author rebuttal**
>
> Thanks to the authors for their efforts. I have read the rebuttal and the comments from the other reviewers, and most of my concerns have been addressed. It's good to see the added disentanglement metrics and the qualitative comparisons. I also recommend to include these in the revised paper. After consideration, I decided to raise my score from 5 to 6.

---

> > ### Author Response · Authors · 2023-08-18
> > **Thanks for raising the score!**
> >
> > Thanks for raising the score and the encouraging feedback!
> >
> > We would definitely include the extra experiments and the results of disentanglement metrics in the revised paper. We sincerely appreciate your constructive suggestions for improving our paper.

---

### Official Review · Reviewer_ww47 · 2023-07-08

**Soundness:** 3 good
**Presentation:** 3 good
**Contribution:** 3 good
**Rating:** 6
**Confidence:** 3

**Summary:**

The authors study disentangled and equivariant representation learning as the authors claim existing approaches are either ill-specified or can not effectively separate realistic factors of interest in the learned latent space. The key idea is to define a distinct set of tangent directions that associate different transformations. This is achieved by using flow factorized variational autoencoders, and encouraging each latent posterior flow to follow the optimal transport path, which is accomplished by using the PINN loss (equation 14) based on the Hamilton-Jacobi equation with an external driving force.

**Strengths:**

The authors provide an alternative viewpoint at the intersection of disentangled and equivariant-based representation learning, and propose the flow factored representation learning which targets to define distinct sets of latent probability paths that explains different input transformations. The authors designed a sequential variational autoencoder with latent posterior flow following the OT path. This is achieved by making the gradient satisfy some PEDEs.

The graphical model, illustration and derivation overall look sound to me (I did not check Theorem 1). The authors also examine their proposed methods in two established datasets (MNIST and Shapes3D). In the two datasets, the proposed methods achieve clear benefits in terms of log-likelihood and equivariance error.

As the proposed method is targeting for learning equivariant representations, it is expected that the method does achieve lowest equivariance error; Meanwhile the latent flow does perform the target transformation precisely during evaluation while leaving other trains unaffected according to the qualitative study.

**Weaknesses:**

The methods study the intersection of disentanglement and equivariance, but quantitative results associated with disentanglement are missing. It would still be interesting to evaluate the proposed methods in terms of some major disentanglement metrics (e.g., MIG,DCI,Modularity) when compared with other existing approaches. Also, it would be a good test to benchmark the proposed methods in some other well-established datasets like dSprites, Natural, Falor3D and Issac3D, to examine equivariance error, some disentanglement metrics, reconstruction, or some other downstream tasks.


**Questions:**

In equation 6,  will t=0 be problematic?

In page 4 (right before equation 9), it’s unclear from the notation if you are defining one Neural potential for each value in the category, or you are using only one MLP.

The external force appeared abruptly in equation 13 without any introduction, and any details on neural parameterization (e.g., what does MLP looks like) are missing. Looking at equation 14, it’s unclear why an independent MLP (the f(z, t)) would affect the potentials (those u^k(z, t) which are also defined by MLP).

According to table 1 and 2, the weakly supervised approach still outperforms other baseline methods (which is good), but are consistently worse than supervised ones (expected). It would be interesting to see if semi-supervised learning could improve over supervised one.

**Limitations:**

Do not identify any negative societal impact.

---

> ### Author Rebuttal · Authors · 2023-08-09
>
> We thank $\textcolor{magenta}{Reviewer\ ww47}$ for the encouraging feedback and valuable suggestions. Below we respond to the concerns in detail.
>
> ---
>
> **Q1. Disentanglement Metrics**
>
> As stated in line29 of the supplementary (and also indicated by $\textcolor{blue}{Reviewer\ XnNs}$), most traditional disentanglement metrics like MIG [a] and DCI [b] do not suit our method because these metrics (assume and) require that each latent dimension is responsible for one semantic and manipulating single dimensions of the latent variable would involve distinct output transformations. However, for the recent disentanglement methods including ours [c,d,e], there emerges a more realistic disengagement setting: all the latent dimensions are perturbed by vectors for meaningful output variations.
>
> Nonetheless, as pointed out by $\textcolor{blue}{Reviewer\ XnNs}$, certain disentanglement metrics such as VP scores [f] can be leveraged as they do not pose any assumptions on the latent space but only require image pairs $\mathbf{x} _0,\mathbf{x} _T$ of different transformations for evaluation. The VP metric adopts the few-shot learning setting (using $1$\% or $10$\% of the dataset as the training set) and takes a light-weighted neural network for learning to classify image pairs $\mathbf{x} _0,\mathbf{x} _T$ of different attributes. The generalization ability (validation accuracy) can be thus regarded as a reasonable surrogate for the disentanglement ability.
>
> Due to the limited space, we put the evaluation results in the response to $\textcolor{blue}{Reviewer\ xnNs}$. Our method still achieves superior performance on the VP scores and outperforms the previous disentanglement baselines.
>
> > [a] Isolating Sources of Disentanglement in Variational Autoencoders. NeurIPS18.
> >
> > [b] A Framework for the Quatitative Evaluation of Disentangled Representations . ICLR18.
> >
> > [c] Topographic vaes learn equivariant capsules. NeurIPS21.
> >
> > [d] WarpedGANSpace: Finding non-linear rbf paths in GAN latent space. ICCV21.
> >
> > [e] Closed-form factorization of latent semantics in gans. CVPR21.
> >
> > [f] Variation Predictability for Disentanglement. ECCV20.
>
> **Q2. Application on Complex Real-world Datasets**
>
> Thanks for the constructive advice! We follow your suggestion and further evaluate our method on Falcor3D and Isaac3D. Due to the limited space, please refer to the response to $\textcolor{red}{Reviewer\ tD2D}$ and the attached one-page pdf for the relevant results and discussions.
>
> **Q3. Eq. (6) with t=0**
>
> Thanks for the careful check! When $t=0$ the KL div of Eq. (6) is indeed a bit problematic as the conditioned posterior $q(z_t|z_{t-1},k)$ starts with $t=1$. The more rigorous form could be obtained by injecting Eq. (4) into the KL div part of Eq. (6):
>
> $$\log p(\bar{\mathbf{x}} | k)\geq \sum _{t=0}^{T} E _{q _{\theta}(\bar{\mathbf{z}}| k)} \big[ \log p(\mathbf{x} _{t}|\mathbf{z} _{t},k)\big] -E _{q _{\theta}(\bar{\mathbf{z}}| k)}\big[\mathrm{D} _{\text{KL}}\left[q _\theta(\mathbf{z} _0|\mathbf{x} _0)||p(\mathbf{z} _0)\right] \big] - \sum _{t=1}^{T} E _{q _{\theta}(\bar{\mathbf{z}}| k)} \big[\mathrm{D} _{\text{KL}} \big[\left[q _\theta(\mathbf{z} _{t}|\mathbf{z} _{t-1}, k) || p(\mathbf{z} _{t} | \mathbf{z} _{t-1}, k)\right]\big]$$
>
> The modification decomposes the second r.h.s.term of Eq. (6) into the initial KL div at $t=0$ and the Kl div of the subsequent sequence $t=1:T$. This is also more coherent to our sequential assumption as the first element is independent of the specific transformation $k$. We will revise Eq. (6) for better clarity.
>
> **Q4. One MLP for One Transformation**
>
> Thanks for the detailed comment. We define a single MLP for each transformation. The MLP $u(\mathbf{z},t)\in R^1$ is a scalar and the gradient $\nabla _\mathbf{z} u(\mathbf{z},t)\in R^d$ takes the same dimensionality as the latent variable. Therefore, the gradient of one MLP can drive the evolution of the entire latent variable. In the revised paper, we would introduce the dimensionality of $u$ and $\nabla u$ in Sec 3.3 to avoid confusion.
>
> **Q5. External Force**
>
> We agree that it might be abrupt to jump to the external force immediately after introducing the generalized HJ equation in Eq. (13). Actually, the external force is set as learnable MLPs simply because we would like to achieve a flexible negativity constraint (in order to satisfy the optimality condition). This can be also achieved by setting $f(z,t)$ as learnable parameters at different timesteps, but we empirically observed that parameterizing $f(z,t)$ as an MLP is more stable throughout the training process. We use the same MLP architectures for both $u(z,t)$ and $f(z,t)$.
>
> We would re-organize this part and explain the external force more clearly in detail in the revised paper.
>
> **Q6. Semi-supervised Learning**
>
> Thanks for the insightful advice. Of course, our method can be extended to the semi-supervised setting. One straightforward implementation could be switching between inferring the transformation index from sequences and directly enforcing the index during the training process. We test such a possibility on MNIST and present the results below:
>
> Equvariance error on MNIST under different supervision settings.
> Supervision|Scaling|Rotation|Coloring|
> |:---:|:---:|:---:|:---:|
> |Weakly|193.84±2.47|157.16±3.24|165.19±2.78|
> |Semi|186.07±2.58|152.87±3.36|160.13±2.82|
> |Full|185.42±2.35|153.54±3.10|158.57±2.95|
>
> As can be seen, this semi-supervised setting has very close and even competitive performance against the supervised setting. We also observed accelerated convergence of this setting compared with the weakly-supervised fashion.
>
> Another interesting semi-supervised learning setting is directly enforcing the index for some of the transformations while inferring the index for others. This setting might be useful when the number of transformations to model is very large. We would add a paragraph in the revised paper to discuss the many intriguing possibilities!
>
> ---

---

> > ### Comment · Reviewer_ww47 · 2023-08-18
> > **Thank you for your rebuttal.**
> >
> > Thank you to the authors for your effort and the additional good results. I believe including the latest results do make the paper with higher quality.
> > I also read the opinions from other fellow reviewers. Some fellow reviewers mentioned comparing with powerful generative models like diffusion models, which I’m sure is urgent for the scope of this paper.
> > After consideration, I prefer to keep my score as it is (as I’m hesitant to raise my score further to 7, which means a high impact and good-to-perfect evaluation).

---

> > > ### Author Response · Authors · 2023-08-18
> > > **Thanks for the positive feedback!**
> > >
> > > Thanks for the appreciations on our rebuttal. We would definitely include the latest experimental results in the revised paper.
> > >
> > > We respect your opinion/score on our paper, and we also agree that for better applicability the subfield of equivariant representation learning should try extending to more powerful generative models such as diffusion models.
> > >
> > > As discussed in the response to $\textcolor{green}{Reviewer\ znKv}$, despite the existing challenges in extending our framework, we would keep investigating this direction in future work!

---

### Official Review · Reviewer_tD2D · 2023-07-26

**Soundness:** 3 good
**Presentation:** 3 good
**Contribution:** 3 good
**Rating:** 6
**Confidence:** 3

**Summary:**

This paper proposes a novel approach to unsupervised representation learning; More precisely, the authors focus on the learning of latent dynamics with the aim to design an equivariant mapping from the observational to the representational space; Authors leverage variational posterior approximation and dynamic optimal transport to do so and evaluate their method on MNIST (augmented with standard transformation - rotation, scaling, ...) and dSprites showing a significant log-likelihood and equivariance error improvement in comparison to SlowVAE and TVAE; In addition, authors highlight the flexibility of the approach by showcasing the ability of the trained VAE to generate coherent dynamics in the pixel space with superposed transformations and transformations shifts;

Whilst the motivation towards learning equivariant models as well as the idea of learning disentangled latent dynamics through variational posterior approximation is not new, here authors propose a novel set of assumptions over the posterior ($q(z_t|z_{t-1},k)$) and use optimal transport to efficiently train the VAE;

**Strengths:**

My expertise in optimal transport being limited, I will mostly focus my evaluation on the novelty of the proposed contributions, the soundness of the ELBO objective proposed as well as the assumptions considered;

- I genuinely enjoyed reading this paper and found the contribution exciting. The manuscript is well-written, mathematical derivations are sound, and the experimental setting is clear; The fact that authors show a competitive performance of the model whilst considering only the information from element $x_0$, the first observations of a sequence, in their posterior factorization is exciting. This structural assumption sounds to me more intuitive and sound than the ones I encountered in previous works.
- Whilst the datasets considered in this paper remain quite simple, as generally observed in this sub-field, the improvements in terms of log-likelihood and equivariance error (standard evaluation metrics) are significant and the model is general enough to allow for nice properties to arise (i.e., superposition of transformation, transformation shift);


**Weaknesses:**

- The weakness of this work lies for me in the lack of diversity in the experimental setup considered as further detailed in the limitations sections.

**Questions:**

- Could authors clarify how the number of transformations, $k$ was chosen in the experimental work and whether an ablation over this hyperparameter was performed?
- Could authors discuss the computational complexity of the proposed approach and provide some insights on how this compares to the baselines considered?


**Limitations:**

- As for most work in this sub-field (c.f., SlowVAE, TVAE), the experimental work is limited to simple datasets (i.e., MNIST, dSprites) and does not investigate how these methods perform on real-world images/videos;

---

> ### Author Rebuttal · Authors · 2023-08-09
>
> We thank $\textcolor{red}{Reviewer\ tD2D}$ for the constructive suggestions and positive feedback. Below we respond to the questions point by point.
>
> ---
>
> **Q1. Choosing the Value of $k$**
>
> For MNIST, we choose the value of $k$ in line with TVAE [a] and PoFlow [b]. For Shapes3D, we use the self-contained transformations of the dataset but omit the transformation of "Object Shape". This is because the sequence is very short (4 elements) and its transition is very abrupt, which does not define a smooth and natural transformation like the other sequences.
>
> Equivariance error of different k on MNIST
> k|Scaling|Rotation|Coloring|
> |:---:|:---:|:---:|:---:|
> |1|185.27±2.59|-|-|
> |2|185.78±2.21|154.29±2.87|-|
> |3|185.42±2.45|153.54±3.10|158.57±2.95|
>
> We did an ablation study on the impact of $k$ on MNIST and present the evaluation results in the above Table. In general, the performance is not affected by the number of transformations being applied. We expect that this is because the transformations are learned by distinct potentials (which are implemented as k different MLPs). Each flow evolves along with the gradient field $\nabla u$ on the potential landscape $u$; _**having multiple latent flows defined on different potential landscapes therefore does not interfere with each other**_.
>
> >[a] Topographic vaes learn equivariant capsules. NeurIPS21.
> >
> >[b] Latent traversals in generative models as potential flows. ICML23.
>
> **Q2. Computational Complexity**
>
> We present the training time cost of our method on MNIST in the Table below, as well as the results of TVAE and PoFlow. In the supervised setting, our method is slightly more time-consuming than PoFlow (around 10%) and than TVAE (around 20%). When it comes to the weakly-supervised fashion, our flow-factorized VAE costs more time, as we need to use the Gumbel-Softmax trick for predicting the transformation index given the whole image sequence ($q(k,\bar{\mathbf{x}})$) and aggregate flows according to the index prediction ($\mathbf{z} _{t+1} = \mathbf{z} _t + \sum _i^K q(k, \bar{\mathbf{x}} ) _i  \nabla u^i$). Nonetheless, we think the overall time cost is still acceptable considering the performance gain over the baselines (especially for the supervised setting).
>
>
>  Training time cost on MNIST each iteration
> ||Ours (weakly-supervised)|Ours (supervised)|PoFlow|TVAE|
> |:---:|:---:|:---:|:---:| :---:|
> |Time (s)|1.076|0.723|0.638|0.591|
>
> **Q3. Application on Real-world Datasets**
>
> Thanks for the great advice! As suggested by $\textcolor{magenta}{Reviewer\ ww47}$, we take a step further to apply our method on Falcol3D and Isaac3D, two complex large-scale and real-world datasets that contain sequences of different transformations. Falcol3D consists of indoor 3D scenes in different lighting conditions and viewpoints, while Isaac3D is a dataset of various robot arm movements in dynamic environments.
>
> Equivariance errors on Falcor3D
> |Methods|Lighting Intensity|Lighting X-dir|Lighting Y-dir|Lighting Z-dir|Camera X-pos|Camera Y-pos|Camera Z-pos|
> |:---:|:---:|:---:|:---:| :---:| :---:| :---:|  :---:|
> |TVAE|11477.81|12568.32|11807.34|11829.33|11539.69|11736.78|11951.45|
> |PoFlow|8312.97|7956.18|8519.39|8871.62|8116.82|8534.91|8994.63|
> |Ours|**5798.42**|**6145.09**|**6334.87**|**6782.84**|**6312.95**|**6513.68**|**6614.27**|
>
>
> Equivariance errors on Issac3D
> |Methods|Robot X-move|Robot Y-move|Camera Height|Object Scale|Lighting Intensity|Lighting Y-dir|Object Color|Wall Color|
> |:---:|:---:|:---:|:---:| :---:| :---:| :---:|  :---:| :---:|
> |TVAE|8441.65|8348.23|8495.31|8251.34|8291.70|8741.07|8456.78|8512.09|
> |PoFlow|6572.19|6489.35|6319.82|6188.59|6517.40|6712.06|7056.98|6343.76|
> |Ours|**3659.72**|**3993.33**|**4170.27**|**4359.78**|**4225.34**|**4019.84**|**5514.97**|**3876.01**|
>
>
>
> The two Tables above present the equivariance errors of our method and TVAE/PoFlow. Notice that the values are much larger due to the increased image resolution. Our method still outperforms other baselines by a large margin and achieves reasonable equivariance error. To visually display the learned latent transformations, we also conducted some qualitative evaluations and put them in the attached one-page pdf (which comprises exemplary latent evolutions of different transformations and exemplary comparisons against baselines). Our method can precisely control the image transformations through our latent flows.
>
> ---
>
> Thanks again and it is more than welcome to ask further questions.

---

> > ### Comment · Reviewer_tD2D · 2023-08-15
> > **Rebuttal Acknowledgment**
> >
> > Thank you to the authors for their additional insights and results in particular for the positive results on more complex datasets which provide a more realistic view of the impact of the method;
> >
> > Could authors clarify how the baseline methods were tuned to obtain results shared in the rebuttal answer tables?
> >
> > After consideration, I decided to keep my score; Whilst the additional results provided by the authors increase the quality of the contribution, I believe the definition for a score of 6 (i.e., "Technically solid, moderate-to-high impact paper, with no major concerns with respect to evaluation, resources, reproducibility, ethical considerations.") matches my perspective on this work;

---

> > > ### Author Response · Authors · 2023-08-15
> > > **Thanks for the positive feedback!**
> > >
> > > Thanks for the positive feedback and the appreciation of the additional insights and results in the rebuttal. We respect your opinion on our paper.
> > >
> > > For the baseline methods, we take the official training codes of TVAE and PoFlow, and use the same VAE architectures for all the baseline methods. We also take the same MLP architectures of PoFlow for parameterizing the potentials $u$. The latent space is divided into 18 capsules each of 18 dimensions for TVAE, and the other two methods are defined with latent spaces of the same sizes ($18\times18=324$).

---

### Author Rebuttal · Authors · 2023-08-09

We sincerely thank all the reviewers for their constructive suggestions and valuable feedback. We appreciate the reviewers' common sentiment that our work is **intuitive and novel** ($\textcolor{red}{R1}$, $\textcolor{magenta}{R2}$, $\textcolor{green}{R4}$), **mathematically sound and technically solid** ($\textcolor{red}{R1}$, $\textcolor{magenta}{R2}$, $\textcolor{blue}{R3}$, $\textcolor{pink}{R5}$), **written and organized well** ($\textcolor{red}{R1}$, $\textcolor{magenta}{R2}$, $\textcolor{blue}{R3}$), and has **significant performance improvements** and **good generalization ability/flexible composability** ($\textcolor{red}{R1}$, $\textcolor{magenta}{R2}$, $\textcolor{blue}{R3}$, $\textcolor{green}{R4}$, $\textcolor{pink}{R5}$). We are also glad that $\textcolor{magenta}{R2}$ and $\textcolor{green}{R4}$ clearly agree that our work provides an **alternatively new viewpoint at the intersection of disentangled and equivariant representation learning**.

We have replied to each reviewer individually to address the concerns. Below we answer the common questions and summarize the main changes in the response.

---

**Q1. Application on Complex Real-world Datasets ($\textcolor{red}{R1}$,$\textcolor{magenta}{R2}$,$\textcolor{green}{R4}$,$\textcolor{pink}{R5}$)**

One main concern of several reviewers is the applicability of our method on real-world datasets, as most works in the field benchmark their methods only on toy datasets. As suggested by $\textcolor{magenta}{R2}$, we take a step further to evaluate our approach on Falcol3D and Isaac3D, two complex real-world datasets that contain sequences of different transformations. Specifically, Falcol3D consists of indoor 3D scenes in different lighting conditions and viewpoints, while Isaac3D is a dataset of various robot arm movements in dynamic environments.

We have presented the quantitative evaluation in the individual responses, and the results demonstrate that our method still outperforms other approaches and achieves reasonable equivariance error. Moreover, to visually display the learned latent transformations, we also conducted some qualitative evaluations and attached them in the attached one-page pdf. Our method can precisely control the image transformations through our latent flows.

**Q2. Disentanglement Metrics ($\textcolor{magenta}{R2}$,$\textcolor{blue}{R3}$)**

As stated in line29 of the supplementary (and also confirmed by $\textcolor{blue}{R3}$), most traditional disentanglement metrics like MIG and DCI do not suit our method because these metrics require that each latent dimension is responsible for one semantic and manipulating single dimensions of the latent variable would involve distinct transformations. However, for the recent disentanglement methods including ours, there emerges a more realistic disengagement setting: all the latent dimensions are perturbed by vectors for semantically meaningful output variations.

Nonetheless, also as pointed out by $\textcolor{blue}{R3}$, certain disentanglement metrics such as VP scores can be leveraged as they do not pose any assumptions of the latent space but only requires image pairs $\mathbf{x} _0,\mathbf{x} _T$ of different transformations for evaluation. The VP metric adopts the few-shot learning setting (using $1$\% or $10$\% of the dataset as the training set) and takes a light-weighted neural network for learning to classify image pairs $\mathbf{x} _0,\mathbf{x} _T$. The generalization ability (validation accuracy) can be thus regarded as a reasonable surrogate for the disentanglement ability.

We adopted the VP metric and evaluated the performance of the baseline methods on MNIST and Shapes3D. Please find the results in the individual responses. The evaluation results demonstrate that our method achieves consistently superior performance on the VP scores under different training set split ratios. This indicates that our flow-factorized VAE has better disentanglement performance.


**Q3. Model Complexity ($\textcolor{red}{R1}$,$\textcolor{pink}{R5}$)**

In the individual responses, we present the training time cost of our method. In the supervised setting, our method is only slightly more time-consuming than PoFlow (around 10%) and TVAE (around 20%). We hope the overall time cost of our method is acceptable considering the performance gain over the baselines.


**Q4. Ablation Study on $k$ ($\textcolor{red}{R1}$,$\textcolor{blue}{R3}$)**

We choose the value of $k$ mainly following the literature and the specific dataset specifications. We conducted an ablation to investigate its impact and find out that the results are not affected by the number of transformations being applied. We expect that this is because the transformations are learned by distinct potentials (which are implemented as k different MLPs). Each flow evolves along with the gradient field $\nabla u$ on the potential landscape $u$; _**having multiple latent flows defined on different potential landscapes therefore does not interfere with each other**_.



---

*For brevity, we refer to reviewers $\textcolor{red}{tD2D}$ as $\textcolor{red}{R1}$, $\textcolor{magenta}{ww47}$ as $\textcolor{magenta}{R2}$, $\textcolor{blue}{xnNs}$ as $\textcolor{blue}{R3}$, $\textcolor{green}{znKv}$ as $\textcolor{green}{R4}$, and $\textcolor{pink}{wDok}$ as $\textcolor{pink}{R5}$, respectively.

---

### Decision · Program_Chairs · 2023-09-21

**Decision:**

Accept (poster)

**Comment:**

The paper received mixed initial scores. The common concerns shared by reviewers include the absence of comparisons on complex datasets, the use of disentanglement metrics and the incorporation of modern generative baselines (such as styleGAN and diffusion models).

The authors provide empirical results on two additional benchmarks, and provide disentanglement comparisons in their rebuttal. The authors’ responses steered reviewers towards a consensus to accept this paper. The final version should include the additional benchmark results and disentanglement comparisons discussed in the rebuttal, as well as more in-depth discussions concerning modern generative models.